# Are Kenya Meteorological Department heavy rainfall advisories useful for forecast-based early action and early preparedness for flooding?

David MacLeod[1,2], Mary Kilavi[3], Emmah Mwangi[4], Maurine Ambani[4], Michael Osunga[4], Joanne Robbins[5], Richard Graham[5], Pedram Rowhani[6], and Martin C. Todd[6]

[1]School of Geographical Sciences, University of Bristol, BS8 1SS, United Kingdom
[2]Atmospheric Oceanic and Planetary Physics, Department of Physics, University of Oxford OX1 3PU, United Kingdom
[3]Kenya Meteorological Department (KMD), Nairobi 00100 GPO, Kenya
[4]Kenya Red Cross Society, Nairobi 00100 GPO, Kenya
[5]Met Office, Exeter EX1 3PB, UK
[6]Department of Geography, University of Sussex, Brighton BN1 9QJ, UK
**Correspondence:** David MacLeod (David.MacLeod@physics.ox.ac.uk)

**Abstract.** Preparedness saves lives. Forecasts can help improve preparedness by triggering early actions as part of pre-defined protocols under the Forecast-based Finance / Action (FbF/A) approach, however it is essential to understand the skill of a forecast before using it as a trigger. In order to support the development of early action protocols over Kenya we evaluate the 33 heavy rainfall advisories (HRA) issued by the Kenya Meteorological Department (KMD) during 2015-2019.

The majority of HRA warn counties which subsequently receive heavy rainfall within the forecast window. We also find a significant improvement in the advisory ability to anticipate flood events over time, with particularly high levels of skill in recent years. For instance actions with a two-week lifetime based on advisories issued in 2015 and 2016 would have failed to anticipate nearly all recorded flood events in that period, whilst actions in 2019 would have anticipated over 70% of the instances of flooding at county level. When compared against the most significant flood events over the period which led to

significant loss of life, all three such periods during 2018 and 2019 were preceded by HRA and in these cases the advisories accurately warned the specific counties for which significant impacts were recorded. By contrast none of the four significant flooding events in 2015-2017 were preceded by advisories. This step-change in skill may be due to developing forecaster experience with synoptic patterns associated with extremes as well as access to new dynamical prediction tools that specifically address extreme event probability - for example, KMD access to the UK Met Office Global Hazard Map was introduced at the

end of 2017

    Overall we find that KMD HRA effectively warn of heavy rainfall and flooding and can be a vital source of information for early preparedness. However a lack of spatial detail on flood impacts and broad probability ranges limit their utility for systematic FbF/A approaches. We conclude with suggestions for making the HRA more useful for FbF/A and outline the developing approach to flood forecasting in Kenya.

# 1 Introduction

Like many worldwide the Kenyan population are at significant risk from heavy rainfall-induced flooding. In the last two years alone flood losses and damages have been extensive. Recent examples of this include flooding during the *'Long Rains'* season of 2018, impacts of which included the displacement of 300,000 people (OCHA, 2018). This was shortly followed by the *'Short Rains'* flooding of 2019 which induced a landslide in West Pokot, killing 72 (reliefweb, 2019). In response to this kind of hydro-meteorological risk the Red Cross Red Crescent movement has pioneered Forecast-based Finance/Action approach (FbF/A, see https://www.forecast-based-financing.org/ for more details).

In the humanitarian action landscape FbF/A sits within a wider set of approaches to anticipatory risk management which can broadly be termed Early Warning-Early Action, of which there are many examples (see Wilkinson et al., 2018, for a review of FbF/A initiatives). FbF/A specifically has three defining features: a set of objective pre-defined forecast triggers, which when met activate a set of pre-defined early actions, themselves funded by a dedicated finance mechanism. Together these constitute the Early Action Protocols (EAPs) of an FbF/A system. The EAPs can facilitate early actions (such as evacuation or cash transfers) or readiness actions (such as pre-positioning of non-food items) which can be implemented before the hazard event occurs, thus moving from disaster response to early preparation and reduction of potential risks posed by the hazard event. Many FbF/A pilots are active worldwide and whilst it is not simple to precisely quantify the impact of such programs, evidence suggests they can significantly reduce individual and community expenses (Gros et al., 2019) along with bringing unquantifiable benefits to lives and livelihoods.

Following the establishment of the DREF (Disaster Risk Emergency Fund) by the International Federation of Red Cross and Red Crescent Societies in December 2017, national Red Cross and Red Crescent societies are working to define their EAPs for the dominant hazard types. In Kenya this work is facilitated through the project "Innovative Approaches in Response Preparedness" (IARP) funded by the IKEA Foundation and implemented by the Kenya Red Cross Society (KRCS) with further support from aligned projects notably the UK-funded NERC/DFID project "Toward Forecast-Based Preparedness Action" (ForPAc, www.forpac.org). ForPAc has been working since 2017 with partners including KMD and KRCS to establish the scientific basis for FbF/A and investigate the development of anticipatory approaches in Kenya for managing flood and drought risk across a range of forecast timescales.

Setting up a FbF/A EAP for a particular hazard begins by identifying priority risks or impacts that can be addressed by anticipatory early action. The next step is to identify the best forecasts to use to trigger early action. In Kenya under the IARP programme this involved exploring a range of potential forecasts that can support anticipation of the priority risks and evaluating the accuracy (or, skill) of the forecasts. Anticipatory actions are then selected which are consistent with the skill of the forecast. For instance a reliable forecast of extremely high probability of imminent flooding might be an appropriate trigger for a higher-cost intervention such as evacuation, whilst a lower probability level (with a higher chance of action in vain) could still be linked to a lower cost or "no-regret" action such as repair of river dykes.

Forecast skill assessment is therefore an essential step in designing a system for FbF/A. In order to be used (in this case by the KRCS and national disaster management agencies) forecasts must show evidence of skill, which should be quantified. In

addition the forecast must be readily available to the actors from the mandated agency for providing weather forecasts (in this case the Kenya Meteorological Department, KMD). Finally the forecast must be provided in such a way to be easily integrated within the EAP.

Through the IARP programme a "menu" of potential forecasts of flood risk has been developed for the Kenya EAPs. In the absence of a Kenya-wide national flood forecast system (Weingärtner et al., 2018) forecasts of rainfall provide the most appropriate proxy. One key potential forecast for heavy rainfall events that could result in flooding is the KMD heavy rainfall advisories (HRA, described in full in Section 2.1). These text-based advisories are issued on an irregular basis by KMD when forecasters' interpretation of current conditions and the output of dynamical atmospheric models point to risk of heavy rainfall. These advisories are made widely available to the public and risk management agencies in relevant counties.

As these heavy rain advisories are issued from the mandated forecasted agency they have high potential to be used in a systematic manner as an FbF/A trigger in flood EAPs. However the skill of these advisories is unknown. In addition they are developed explicitly for heavy rainfall warnings and only implicitly warn of flooding. Here then we assess the accuracy of the historically issued KMD HRAs and evaluate their potential to be used as a trigger in a FbF/A system for flooding. Understanding the level of skill of the advisories supports the development of early action protocols by disaster managers.

The verification of the advisories also helps to build confidence in early warnings from subjective forecasts. Many forecasts of natural hazards are produced with some level of expert judgement but this subjectivity makes verification difficult as a large number of forecasts produced using a consistent method are rarely available for objective evaluation. Without this evaluation, trust in the forecast producer alone determines confidence in the forecasts. However when a reasonable archive of forecasts is available forecast verification can both help to build confidence in the use of the forecasts and to increase trust in the forecast producer.

The forecast and verification data are described in the following section, along with an outline of the challenges to verification posed by the format of the advisories and the approach taken to meet this challenge. Results follow and the paper concludes with a discussion of the main findings, limitations to the analysis along with recommendations for design and operation of the Kenya EAPs and further research.

## 2 Data and verification approach

### 2.1 Production of the KMD heavy rainfall advisories

The first HRA was issued at KMD on 2nd June 2015 after being introduced as a forecast product as part of the Severe Weather Forecasting Demonstration Project (SWFDP) for Eastern Africa (https://www.wmo.int/pages/prog/www/swfdp/SWFDP-EA. html). This project was implemented with support from the World Meteorological Organisation with the aim of improving the ability of National Meteorological and Hydrological Services (NMHS) to forecast severe weather events, improve the lead time of early warnings as well as the interaction of NMHS with disaster managers before and during the event. The intended audiences for these advisories are national and county risk management agencies, humanitarian organisations, relevant ministries and the media for dissemination to the general public within areas of concern.

The decision to issue an advisory is subjective, informed by dynamical model output and forecaster experience. Every day forecasters at KMD's Severe Weather Forecasting section review forecast products from Global Producing Centres (such as ECMWF, NCEP, UK Met Office and Meteo France) using their judgement to produce a five-day running severe weather forecast. This five-day severe weather forecast is based on areas expected to receive any of the following: rainfall above 50mm in 24 hours, winds greater than 25 knots or waves above 2m height. These forecasts are presented graphically as polygons, along with tables showing the level of risk (low, medium or high) over specified areas. At 0900Z representatives from the NMHS of all the contributing countries of the SWFDP participate in a teleconference call to discuss the forecast and develop a consensus.

If any models indicate a raised chance of an extreme event occurring over Kenya during the next few days then a high impact weather conference is held at KMD by experts from the forecasting unit and a consensus advisory is drafted. A subjective probability of occurrence is estimated based on the consensus between models, taking into account weighting of the better-performing models (where model quality is judged subjectively, according to forecasters' experience). Once the advisory is drafted it is examined and reviewed by the senior management within the forecasting division and finally sent to the Director for approval to disseminate it to the public by the public weather service section.

HRA are the most frequently issued type of advisory by KMD (advisories for strong winds, marine and temperature are also issued but are not considered in this study). The advisories are text-based (an example is shown in figure 1). They generally specify a rainfall threshold which could be reached: sometimes this is included as a rainfall rate (e.g. 30mm in 24 hours), otherwise an accumulation total without a rate is mentioned. Finer scale details are often included in this description, such as when within the valid period the rainfall can be expected to start for different regions. Following the forecast description the full list of potentially affected counties is listed, along with general instructions for flood preparedness (e.g. "be on the lookout for potential floods", "avoid driving through or walking in moving water", "people in landslide prone areas...should be on high alert").

There are no clear objective criteria triggering issuance of HRA, which is a subjective process, depending on forecasters' experience and perception of model skill, consensus within the forecasting section and forecast data available. The forecast information used at KMD to produce the HRA has changed over the advisory period under study: in mid-2016, KMD was granted a two year trial license to ECMWF *'eccharts'* through the SWFDP and since August 2017 KMD began using the UK Met Office Global Hazard Map (GHM) as part of the ForPAc project. The GHM provides an at-a-glance summary of forecast high-impact weather over the coming week, by visualising forecasts from the UK Met Office (MOGREPS-G) and ECMWF (the ENS), both separately and in a multi-model ensemble forecast. The multi-model informs summary polygons which direct forecasters to the potential for high-impact weather over the week ahead, via an overview map.

By the end of 2019 a total of 33 were HRA had been issued. These 33 have been digitised here for the purpose of verification, with relevant information extracted: the date of issue and validity, the probability range, the rainfall threshold specified, along with all counties mentioned. Details are given in table 1 and descriptive statistics are shown in figure 2. Several aspects of the KMD advisories demand a careful approach to verification as detailed in the following section.

## 2.2  Verification approach

There are three characteristics of the HRA with implications for verifying them against observed rainfall:

1. The small sample size (33) means it is difficult to assess specific aspects of the forecast such as reliability of probabilities or accuracy of rainfall thresholds. Descriptive statistics for these are provided in figure 2 which show that the probability range of "33-66%" is indicated in nearly all advisories (figure 2d, used in 26 advisories) and other probability ranges are rarely used.

2. The forecast window over which advisories are active is variable, from one to six days but most commonly out to three days (figure 2c, 13 advisories), so the definition of heavy rainfall for verification cannot be consistent.

3. The spatial characteristics of the forecasted heavy rainfall are ambiguous. To illustrate: should we deem an advisory warning of 50mm of rainfall for two named counties to be a 'hit' if 50mm accumulated rainfall is observed (a) over a single point within at least one of the counties or (b) over the entirety of either or both counties or (c) any areal extent between these extremes? This spatial aspect is further complicated by the wide range of size of Kenyan counties: from just over 200km2 (Mombasa) to over 70,000km2 (Turkana). The hit rate and false alarm rate would be highly sensitive to these verification criteria.

In order to address these issues, we take a step back and refocus on the question: would these advisories have been worthwhile for flood preparedness? Though 'heavy rainfall' does not necessarily lead to flooding, and flooding does not always require a heavy rainfall event for triggering (Berghuijs et al., 2019), we proceed by considering the perspective of a manager responsible for flood preparedness at KRCS who is interested in the consequences of using the advisories as a trigger for preparedness.

We first assume that every advisory triggers preparedness actions, independent of the rainfall threshold or probability specified. We then define the extent of the preparedness actions according to the counties mentioned in the advisory. Such actions are unspecified here and could range from a low-regret communication to county-level Red Cross volunteers to a more expensive decision to pre-position supplies. This approach is consistent with the FbF/A approach, though with action triggered on the release of an advisory rather than being associated with a particular probability level.

After assuming that action was taken within the entire region under advisory for each advisory window, we then consider the question, was this action worthwhile? There is no single answer to this question, as it depends on the specific actions along with individual and institutional tolerances for false alarms and misses. However following this approach we can identify clear hits and false alarms, and can confront the advisories with 'what really happened'. As such our method involves answering the following four questions:

1. How well does the total area under advisory warn of the extent of heavy rainfall? (Section 3.1)

2. What is the relative spatial extent of preparedness actions implied by each advisory? (Section 3.2)

3. How many flooding events in the period 2015-2019 would the advisories have anticipated? (Section 3.3)

4. How often would an FbF/A system based on the advisories be expected to trigger? (Section 3.4)

By answering these questions we determine the extent to which the KMD HRAs could effectively guide preparedness activity.

### 2.2.1 Comparing advisory areas with subsequent rainfall

We address question one with a visual comparison of the total area warned under each advisory with the total rainfall accumulation in the subsequent advisory window. Rainfall observations are taken from the Climate Hazards and Infra-Red Precipitation Data with Stations (CHIRPS) dataset (Funk et al., 2015). We use CHIRPS as it compares favourably against other rainfall datasets over East Africa and benefits from relatively high station density in Kenya (Dinku et al., 2018). Particular weaknesses of CHIRPS include spurious drizzle and underestimation of peak magnitudes of the most extreme rainfall (specifically the 99.9th percentile, Beck et al., 2017), but our focus on multi-day accumulation of heavy but not necessarily extreme rainfall should be insensitive to these biases.

With this visual comparison we begin with a subjective assessment of the overall performance of advisories. Following this we calculate the distribution of accumulation totals across all $5km$ CHIRPS gridpoints inside the polygon associated with the warned counties, quantifying the spatial extent of high rainfall totals for areas under advisory. In addition we show the distribution as the percentage of grid points within the warned region receiving more than a specified rainfall threshold. Throughout the analysis we evaluate the total rainfall accumulation across each variable-length advisory window.

In addition we derive the proportion of the warned area that experienced accumulated rainfall above indicative thresholds. No single rainfall threshold leads to increased flood risk, which depends on many factors both hydrometeorological and social. Even for a single location the same amount of rainfall may cause a flood in one year but not the next. In the following analysis we show results for 25, 50, 75 and 100mm accumulation over the advisory window and focus the discussion on results for 50mm accumulation. We do not suggest that this threshold has primacy over others; an in-depth analysis would be necessary to determine and quantify the most relevant thresholds for flood risk in a location. Instead we take 50mm as a working definition of heavy rainfall to keep the discussion concise, whilst including other thresholds in the analysis for reference.

### 2.2.2 Estimating the relative extent of preparedness actions implied by advisories

To answer question two we estimate the relative scale of preparedness implied by each advisory. In practice preparedness actions would be determined by overlaying the forecast hazard footprint with data on exposure and vulnerability to that hazard. Many different actions are possible which would target different groups and we do not attempt to evaluate the cost of specific actions. Instead we aim at a broad indication of the magnitude of the general preparedness activities appropriate for each advisory, by assuming that preparedness is taken based on advisories to target communities exposed to a one in five year riverine flood event.

We use ward-level exposure data provided by KRCS, which has been created by combining population density with an estimate of the areas inundated by a one in five year flood which has been provided to KRCS by ECMWF and calculated using

the modelling framework of the Global Flood Awareness System (GloFAS). The exposure estimate is not intended to quantify the absolute level of assistance required (not least because the frequency of advisory issuance means that the vast majority will not be followed by a one in five year event by definition). However it does allow a relative estimate of the extent of preparedness action required between advisories. For instance an advisory active in locations where 2 million people are exposed to flooding is likely to require more preparedness than an advisory relevant for only 200,000 people. It should also be noted that the number exposed to flooding is an upper bound on those actually requiring assistance, as we do not take vulnerability to flooding into account.

We then assess the amount of rainfall falling in the specific areas where people are exposed to flooding and estimate the percentage of the 'prepared people' who received above threshold rainfall. From this we can estimate the relative 'worthiness' of each preparedness action: assuming that when flood preparedness assistance is given in a location and significant rainfall follows the action is considered worthy (even if that heavy rainfall does not lead to flooding). We note the potential mismatch between local rainfall and flooding (e.g. when rainfall falls upstream in catchment and floods lower reaches), which suggests that our assumption of worthiness only when heavy rainfall is experienced locally should be considered a lower bound; inclusion of flooding related to non-local rainfall would only increase the estimate of worthiness.

### 2.2.3 Verifying HRA against flood events and evaluating frequency of action triggering

The analysis so far quantifies the extent of rainfall accumulations and estimates the relative scale of the actions which each advisory may trigger. Whilst heavy rainfall is not the only factor in flooding (Amoako and Frimpong Boamah, 2015) and does not always trigger flooding, flood risk and response managers may be inclined to use the HRA to trigger readiness activities for flooding. It is therefore instructive to verify the issued HRA directly against recorded flood events, answering question three above. We use two sources of flood records and their use in verifying the advisories is described below.

The first flood record database has been created by KRCS. This comprises a county-level record of flood events based on information from the KRCS Emergency Operations Center(EOC). The EOC operates 24 hours a day at KRCS headquarters and records disaster incidence that are recorded all over the country on social and mainstream media and by KRCS volunteers. The record from the EOC has been supplemented with additional events identified *post hoc* from other online sources. In total over the five years 2015-2019 the database notes 461 flood events, with 167, 44, 54, 164 and 199 for each year separately (NB simultaneous flooding in two counties is considered in this count as two events).

The KRCS flood record is then used to calculate two key skill statistics across the entire sample (over all counties). Firstly the hit rate (HR), calculated here as the percentage of events which were preceded by advisories. Secondly we calculate the precision, which is defined as the percentage of advisories which are followed by a flood event (NB precision is equal to 100% minus the false alarm ratio, another key metric for FbF/A, and is a commonly-used diagnostic in informatics Powers (2011)). HR and precision are calculated over the whole sample and for each year separately. Following Coughlan de Perez et al. (2016) they are also calculated under the assumption that actions related to flood preparedness have a lifetime, that is, preparedness carried out today will still avert flood risk even if that flooding does not occur immediately. Actions such as evacuation will only remain effective whilst people remain evacuated, whilst low-regret actions focused on readiness such as pre-positioning

of water purification tablets will still be useful if flooding occurs months later. Coughlan de Perez et al. (2016) use a 30 day lifetime in their verification; here we evaluate the advisories across a range of action lifetimes from 0 to 30 days following the end of the advisory window.

The second source of flood record we use is the EM-DAT database (EM-DAT, 2020). EM-DAT collects data on the occurrence and effects of mass disasters globally, requiring at least one of the following four conditions for inclusion in the database:

- 10 or more people dead;

- 100 or more people affected;

- The declaration of a state of emergency

- A call for international assistance

Eight significant flood events in Kenya are recorded in EM-DAT for the period June 2015 to December 2019. From these we remove the Solai earth dam collapse of May 2018 as there were major non-meteorological reasons for its collapse (including lack of maintenance and an outdated design, NECC, 2018). We merge the two entries beginning 14th March 2018 as they relate to the same period of heavy rainfall. This leaves six flood events, to which we add the landslide of November 2019, as this was directly triggered by a period of heavy rainfall. Compared to the KRCS record, the EM-DAT record is much smaller and so precludes a robust quantitative analysis. Instead we consider each event in turn and determine the relevance of the advisories for anticipating these most significant flooding events, for which early warning would have been most valuable.

Finally we conclude by addressing question four. Here we determe the number of times a FbF/A system based on HRAs might be expected to trigger in each county. We assume here that actions have a lifetime as described above, and that the system will not be triggered again if an action is still active in that county.

## 3 Results

### 3.1 How much rain fell in counties under HRA?

We begin by identifying the total area of all counties named in each HRA and compare this with the accumulated rainfall over Kenya during the advisory valid window. For convenience, advisories are labelled (A-Z, followed by A' to G') in table 1 and these labels are used from this point.

Figure 3 shows all the advisories and the resultant accumulation. From a visual comparison, we see that eighteen advisories provide a good forecast of all areas going on to receive at least 50mm rainfall accumulation (A, F, H, J, K, L, P, R, S, Y, Z, A', B', C', D', E', F' and G'). For these advisories preparedness is most likely to have been considered worthy, and local actions based on these advisories are likely to be hits.

Nine advisories do successfully warn of heavy rainfall in some areas, whilst failing to warn other counties which received similar amounts (G, I, M, N, O, T, V, W and X). In these cases preparedness may have been considered worthy, although

preparedness would not have reached all those potentially affected by flooding, with risk of missed events and therefore failing to act.

Five advisories warned the "wrong" counties, where more accumulation was seen in unwarned counties than those receiving warnings (C, D, E, Q and U). One advisory (B) warned coastal counties of heavy rain yet 20mm fell during a two-day window, a relatively normal amount for the region. For these six advisories it is unlikely that preparedness triggered by the advisories would be considered worthwhile, instead would possibly be seen as false alarms and misses.

Next, we consider the rainfall distribution across these regions under advisory. Figure 4(a) shows the rainfall accumulation across the warned region for each advisory, presented as the distribution over the sample of $25km^2$ CHIRPS gridpoints. Figure 4(b) shows the percentage of the warned area which receives rainfall accumulation above thresholds 25, 50, 75 and 100mm. We see that for the vast majority of advisories (29 out of 33), less than 50% of the warned area received over 50mm. This implies that for any point location falling in an area under advisory there is generally over 50% chance that no 'significant' accumulation will be seen. This is inevitable for rainfall early warnings, particularly in a region with a large contribution from localised but intense convective storms, leading to high spatial variability in rainfall totals.

From a meteorological perspective then we find the advisories to be relatively good indications of heavy rainfall. Summarizing the above semi-quantitative analysis of figures 3 and 4, we conclude that 18 successfully warned those regions which did receive heavy rainfall, nine provide warning for some regions but miss other regions, whilst only six of 33 are unlikely to be useful for early preparedness actions. However at the same time, nearly all 'good' advisories warn significantly larger areas compared to the areas which go on to receive heavy rainfall.

We next turn to potential actions triggered by the advisories; estimating the relative extent of preparedness action implied by advisories along with the potential public perception of the actions based on locally experienced rainfall.

## 3.2   What is the extent of preparedness action implied by advisories?

Ward-level density of the population exposed to one in five year flooding is shown in figure 5. High population density is seen around the Lake Victoria basin and elsewhere in the central highlands, although large areas of this highly-populated region are not exposed to significant flood risk. This indicates the importance of taking patterns of exposure into account. This population density is then integrated across the warned region for each advisory to estimate the total number of exposed people warned by the advisory. This is shown as the black stars in figure 6(a).

Significant variability in the extent of the warnings for the population at risk from flooding: eight advisories warn areas where at least one million people are exposed to flooding. The rest warn around 500,000 people and fewer, and of these the warning from 18 advisories is 'only' targeted at fewer than 200,000 people (these smallest scale warnings are generally when only warnings for coastal counties are active). This quantifies the significant variations in the extent and cost of preparedness actions which could be linked to the advisories.

To evaluate the extent to which this preparedness would have been perceived as worthwhile, we also show the number of exposed people living in a warned area which then went on to receive accumulation of 25, 50, 75 or 100mm. These results are also shown in figure 6(a), whilst figure 6(b) presents these values as a percentage of the population warned which received rain-

fall above each threshold. Since these scores are conditioned on exposed population, they are highly sensitive to the underlying exposed population density. They will only be improved if heavy rain falls on an area at risk from flooding, and this improvement will be higher if the area is more densely populated. In this way we move beyond purely meteorological verification and take into account real-world implications of acting on a forecast. This also considers the potential response of beneficiaries of flood preparedness: if flood preparedness is carried out in a region that subsequently receives significant rainfall, most people will see the preparedness as worthwhile. Conversely, people are more likely to see the action as a false alarm if no significant rainfall falls where they live.

Focusing again on 50mm accumulation as a nominal threshold for increased flood risk, we see several advisories for which most people receiving early preparedness would not have seen significant rainfall. For eight advisories less than 10% of those receiving assistance would have seen more than 50mm; these are unlikely to be seen by most as worthy actions (A-E, P, Q and U). At the other end of the scale, six advisories see significant accumulation for at least 60% of those assisted (M, T, X, A', C' and E'). The remaining 24 see significant rainfall for between 10-40% of those affected. Notably by this metric the first five advisories (covering mid 2015 to mid 2017) are among the worst-performing, whilst those most likely to have led to worthy actions were all issued in 2018 and 2019.

### 3.3   Did advisories warn of flooding?

We next turn to the verification of the advisories against recorded flooding in the KRCS flood record. HR and precision are shown in figure 7. This shows a clear improvement of the advisories over time: for advisories in 2015 and 2016 less than 5% of flood events were hit, even with a favourable assumption of 30 day lifetime of preparedness actions. Conversely action on advisories in 2019 would have seen a 40% HR with a zero day lead time, rising to 60% or over 70% if actions are taken with a one or two week lifetime. Though 2019 also saw many more advisories issued compared to earlier years, each was also more precise, with a 40% chance of seeing flooding in a county within two weeks of taking action during 2018 and 2019, compared with 20% in 2017, 10% in 2016 and 0% in 2015.

Though recent advisories perform well when measured against the KRCS record of flooding, it may not be that all events in the record would have required significant preparedness. We therefore turn now to the seven most significant flooding events in Kenya over the period, recorded in the EM-DAT database. These are compared with relevant advisories; for simplicity we consider an advisory to be relevant if it was issued in the seven days preceding the indicated start date of the impact, as early preparation triggered by that advisory would have been in place for the onset of the event. We do not require the heavy rainfall window to explicitly overlap with the recorded period of impact, allowing for some lag between heavy rain and flooding. The locations and details of the events are plotted in figure 8 which also shows the counties mentioned in any relevant advisories as defined above (if any). These seven events are now discussed in turn.

Figure 8a shows the significant flooding which occurred across Kenya in December 2015 during the large 2015 El Niño event that peaked in December. This event led to the highest number of deaths recorded in the sample (112). No HRA was issued at any point before or during this event, or during the season as a whole. Notably seasonal forecasts did indicate an

increased risk of a particularly wet season; although as a whole the seasonal rainfall anomalies were smaller than previous comparable El Niño events (Siderius et al., 2018; MacLeod and Caminade, 2019).

Figure 8b represents a smaller event in Turkana county caused by intense rainfall on a single afternoon (10th March 2016). This rainfall led to river overflow, three deaths, displacement of 1,000 people and loss of livestock. No HRA was issued for
this event.

The third event (figure 8c) occurred at the end of April 2016. This flooding impacted over 10,000 people across semi-arid counties in the north (Turkana, Marsabit and Wajir) along with Nairobi. In Nairobi the rainfall triggered the collapse of a building in the Huruma estate (a building which was not constructed to safe standards) ultimately leading to 52 deaths. In advance of this period a HRA was issued by KMD (advisory C), however warnings were given for coastal counties and parts of
Western Kenya but not for those counties most seriously impacted. KRCS did trigger an early response based on this advisory, activating response teams and sending out warnings via SMS to communities living in lowland areas. Although no heavy rainfall was directly experienced in those regions for which the response was triggered, the action was felt to be worthwhile at KRCS as some flooding was experienced later due to Tana River bursting its banks after heavy rainfall in the central highlands.

The next EM-DAT event occurred in May 2017 (figure 8d). This involved coastal counties along with some in the central
highlands and some in the west. 26 deaths were recorded with over 25,000 affected for this event, during which a reported 235mm of rain fell on Mombasa in a 24 hour period between 8-9 May. Although an advisory for coastal counties was issued in late April (advisory E), the valid period was a single day which saw little accumulation in the warned counties.

Figure 8e shows the impacts of heavy rainfall during the 2018 long rains season, which has been evaluated in depth elsewhere (Kilavi et al., 2018; Finney et al., 2019). Widespread flood impacts were seen across the country beginning on 14 March and
extending throughout the month. Two advisories were issued during March (advisories K and L). The first was issued on the 9th and covered the period 13-15th and a follow-up was issued on the 15th, covering the period 16-19th. Both of these periods saw significant rainfall accumulation (see figure 3 and Kilavi et al., 2018). Every county noted in EM-DAT as experiencing flood impacts was mentioned in these advisories, except for Mandera in the extreme northeast of Kenya.

Figure 8f shows impacts occurred from 17-24 October during the short rains 2019. Flash floods, landslides and riverine floods
were reported in Turkana, Wajir and Elgeyo-Marakwet counties. Two advisories were issued preceding this event (advisories Z and A'). The first was issued on the 10th, covering the period 10-14th and a second was issued on the 14th, covering the period 16-20th. Counties with reported flood impacts were all mentioned in these HRAs.

The final event in the sample also occurred during the 2019 short rains: a landslide in West Pokot on the 23rd November (figure 8g). This occurred following heavy rainfall across many counties, for which a warning was issued several days ahead
of the event on the 18th November, covering the 19-24th of the month (advisory C').

In summary the first four events in the study period were not well warned by advisories. The third event in April 2016 was preceded by a warning but it did not target the counties with significant flood impacts. The final three events in 2018 and 2019 were all preceded by advisories correctly targeting the counties which saw major impacts from heavy rainfall; the lead time between the first advisory and the recorded start of the impacts for these three events was five, seven and five days respectively.
Advisories issued in 2018-2019 therefore gave effective warning to areas experiencing significant flooding impacts, whilst

the earlier advisories did not. Along with skill analysis shown in figure 7 this suggests that in recent years advisories have improved, and have the potential to act as a trigger for an FbF/A system. However it should be recalled that the warned area is often much larger than the area experiencing heavy rainfall (see figures 4, 6, 8). Even those advisories where triggering leads to worthy action where impacts are felt will also simultaneously trigger action in many places which do not require early preparedness, and these 'actions in vain' may be quite expensive in highly populated regions such as West Kenya. In the next and final section, we turn to a practical consideration of basing such a system on advisories and estimate how often such a system might be expected to trigger.

## 3.4 How often would an FbF/A system based on advisories trigger?

An important consideration in setting up an FbF/A system is how frequently it can be expected to be activated. It is desirable to prepare for all significant events, however more frequent triggering limits the cost of actions if the system is to remain financially sustainable. Here we estimate how often such a system might trigger.

Naturally the number of advisories will fluctuate year to year depending on climate variability. However 2018 and 2019 could reasonably indicate the potential number of activations of a FbF/A system, given that they both experienced significant rainy seasons (with 11 advisories issued in 2018 and 13 in 2019, figure 2a). For low-cost actions such as targeted communication of the warning to vulnerable communities this may be an acceptable number of triggers and results from section 3.3 suggest that these would successfully warn against all significant flood events. A key requirement of the advisories is to warn the vulnerable public of significant hazards and so for this purpose the frequency of issuance is appropriate to the cost of the warning.

In the FbF/A context the advisories could be used to instigate actions from response organizations and disaster management. Several actions have already been identified as potentially forming part of an EAP (Maurine Ambani, personal communication):

- Strengthening of barriers designed to prevent people from crossing rivers or places where there is usually fast flowing water

- Provision of water containers and water treatment

- Provision of vouchers to affected populations to access water treatment tablets, containers and treated mosquitoes nets

These kinds of actions would have significant costs and so more than ten triggers in a year may not be realistic. However on the other hand triggering on every advisory may not be necessary. Frequently an advisory is issued which follows on from another, describing a continuing rainfall event (e.g. J-L, M-O, C'-G'). Significant flood preparedness may not need to be carried out for each individual one of the advisories in sequence as actions of this nature will have a "lifetime" that may span the interval between several consecutive issued warnings (Coughlan de Perez et al., 2016). For example river defences will still be effective several weeks after action is taken to repair or reinforce them.

The impact of action lifetime on trigger frequency is illustrated for each county in 2019 in figure 9. Here we assume that the action will not be repeated if another advisory follows closely after the action is triggered. The number of total actions is shown, assuming an action lifetime of one, two, three or four weeks. We consider multiple chained advisories such as C'-G' as

triggering a single preparedness action: after the first days of heavy rain, activity will have already moved from preparedness to response mode, additional advisories may trigger scaling-up of existing response operations.

With an action lifetime of one week most counties would have triggered four times in 2019. With a longer lifetime the system activates less often and in the longest case of four weeks no county would have activated in 2019 more than twice (on average, once for each of the rainy seasons).

Typical FbF/A approaches tend to focus on extreme events rather than floods which occur every year RCRCCC (2020) and so even taking into account long action lifetimes this trigger frequency may still be too high for high cost actions. However

this frequency may yet be appropriate for FbF/A linked to low-cost low-regret actions, such as fast-tracking drainage clearance which has already been planned and budgeted for.

## 4   Discussion and recommendations

Here we have evaluated the KMD HRAs. This has been done from the perspective of a humanitarian agency such as KRCS, as if the advisories were used to initiate a preparedness protocol such as FbF/A in order to reduce risks related to heavy rainfall.

Such EAPs for a national flood FbF/A system are currently being developed. Our assessment of the advisories has considered:

- the relationship between area warned and the subsequent rainfall received

- the scale of preparedness triggered by the advisories and the perception of the actions based on locally experienced rainfall

- whether the most significant recent flood events followed HRAs

- how frequently an FbF/A system could be expected to trigger

We now draw some general conclusions and provide some recommendations for improvement of the HRAs and outline the development of flood risk forecasting in Kenya.

### 4.1   Conclusions

Advisories issued in the 'early period' (from the first in 2015 through to 2017 inclusive) do not appear to be particularly

effective for preparedness for flood or heavy rain impacts. For each of the nine advisories that were issued in this early period the counties which were warned did not generally receive significant amounts of rainfall. Furthermore four significant flood events were reported in this period and none were anticipated by any advisory, whilst 0%, 5% and less than 20% of all recorded flooding of any magnitudes was preceded by advisories in each of 2015-2017, respectively. We conclude then that it is unlikely that conducting preparedness actions based on advisories between 2015-2017 would have effectively reduced flood or heavy

rain impacts.

However we note evidence of an improvement in the potential utility of advisories in recent years of 2018 and 2019, where they were more frequently issued. Notably these years had particularly wet seasons, March-May 2018 and October-December

2019. For a two week action lifetime, preparedness at county level based on advisories in 2018 and 2019 would have anticipated 40% and 70% of all 363 recorded county-level flooding in these years, whilst the three periods which saw significant mortality directly associated with heavy rainfall which were well-warned by advisories. We conclude then that advisories issued across 2018-19 were particularly skillful at anticipating heavy rainfall, and that preparedness actions based on these could have led to reductions in the impacts of the worst floods in this period. If the performance of advisories over this period is indicative of future performance, then they have the potential to effectively anticipate significant flooding impacts in Kenya.

One factor for the improved hit rate in 2018 and 2019 may be the higher frequency of issuance. However this does not explain the fact that infrequent early advisories were not generally followed by significant rainfall as noted above. This poor performance in the early period might instead be related to the novelty of the system. The first advisories were issued in 2015 and it may have taken some time to develop the systems and expertise and gain confidence in issuing advisories. Another explanation for the change in skill is the evolving access to forecast information from global models at KMD.

In mid-2016 KMD was granted a two year trial license to ECMWF *'eccharts'* through the SWFDP which is reported to have been crucial in informing the advisories released during that period (Mary Kilavi, personal communication), and particularly so during the long rains 2018 (advisories J-Q). In addition the GHM in use since August 2017 has provided a multi-model easy-to-interpret visualization of potential severe weather. Evaluation has shown that multi-model forecasts outperform individual models for extreme precipitation (Robbins and Titley, 2018). The availability of a higher skill multi-model forecast at KMD in an easy-to-interpret format may then be a factor in the significant improvement in skill of advisories during 2018 and 2019. Indeed it is reported that the GHM was a key source of information for the advisories which were issued in advance of all three significant heavy rainfall impacts reported during 2018 and 2019 (figure 7e-g). See also Kilavi et al. (2018) for skill analysis of the GHM forecasts use during the 2018 *'Long rains'*.

Overall we demonstrate here in the first systematic verification conducted of the HRA that they have skill. We find an increase in skill over time and that the HRA anticipated the most significant flood events during 2018 and 2019. However we also find they lack spatial precision on the precise location of heavy rainfall impacts which may limit their use as a trigger in KRCS EAPs.

## 4.2 Recommendations

Though the HRA have skill, their likely utility will clearly depend on the specific context of use. In order to fully ascertain appropriate actions which could be triggered by the HRA, an econometric analysis of the costs and avoided losses of a range of preparedness actions is necessary (and recommended). We note here however that their intended purpose is to alert county governments, other agencies and the general public of the possibility of heavy rainfall. For this purpose they are effective: they are widely disseminated, the text identification of counties under advisory requires no technical knowledge to understand and most importantly, they have skill. Indeed, Kilavi et al. (2018) note dissemination and use of HRA during the *Long Rains* 2018.

As a source of information for a systematic FbF/A system for flooding the advisories have several useful characteristics for KRCS: they are produced by the national mandated agency for weather forecasting, they are readily available at no cost and being text-based, they require no specific knowledge for interpretation. However it is likely that they are not suitable for

triggering a KRCS EAP for flood. The county-scale warning limits the spatial precision of interventions and the frequency of the triggering per county is likely to be too high for FbF/A, which is intended to target extreme events with a return period of one in five years or greater. In addition the HRA only provides a general picture of potential flood impacts without taking into account any local hydrological conditions. However given the clear skill of HRA found here there is clear scope of KMD to develop these in the context of Impact-based Forecasting (WMO, 2015): here we make some recommendations for improving the HRAs and the flood forecasting from the perspective of stakeholders such as KRCS.

### 4.2.1 Developing the HRAs

Improvement of the probabilistic information in the HRA would make them more fit for the purpose of FbF/A. A single category 33-66% is issued in nearly all advisories which limits options for preparedness actions. More diverse and precise probabilities would allow a range of increasing levels of preparedness activities, where high-cost actions are only triggered for the highest probabilities. Of course it is essential that these probabilities are reliable, and a relatively low frequency of subjectively developed forecasts makes this aspect of the forecast difficult to evaluate. However the use of historical forecasts and hindcasts from ensemble forecasting systems used in the GHM (Robbins and Titley, 2018) currently in use at KMD would help to establish the reliability of probabilities and provide a scientific basis for issuing more specific heavy rainfall probability forecasts. Analysis of these dynamical models should also evaluate their performance for the four flooding events in the early period of the KMD advisories (figures 7a-d) to see if these systems did capture these events.

The heavy rain warning area could also be more precise by providing it as a free-shape rather than administrative county boundaries. Whilst naming counties in the advisory is essential for communication to the public and to county government disaster risk management structures the precise area of heavy rainfall areas will not align with administrative boundaries and so warning whole counties will tend to overestimate the total area expected to experience rainfall. Such warning polygons are generated by the GHM and forecasts could be based upon this. KRCS could then overlay these with maps of population exposure and vulnerability to flood risk in order to further narrow down targets for intervention. This would then provide the building blocks of an Impact-based Forecasting system, following WMO guidelines WMO (2015).

Finally many preparedness actions are limited by the lead-time of the HRA. They are often issued in the morning of or the day before the expected start to the rainfall, leaving a small window to coordinate and implement preparedness. A longer lead heavy rainfall forecast would extend the scope of preparedness actions. Currently the time afforded by existing 7- and 5-day forecasts from KMD could be used by KRCS to prepare higher-cost actions, which are finally triggered upon the issuance of a HRA for the next few days. This approach would be analogous to the ready-set-go approach of the Red Cross designed to integrate seasonal forecasts into decision making, adapted to a much shorter overall anticipation window (Bazo et al., 2019).

However the provision of forecasts at even longer lead-time could further enlarge the window for preparedness. For instance, subseasonal forecasts have been shown to have skill out to several weeks ahead (Vitart et al., 2017) and there is clear potential for warnings on this timescale to inform humanitarian preparedness (White et al., 2017). Evaluation of these timescales is being carried out as part of the ForPAc project which has identified potential utility over Kenya and these subseasonal forecasts are currently being trialed at KMD after being made available in real-time as part of phase two of the S2S project (Kilavi

et al., 2018; MacLeod et al., submitted). The longer lead time of these rainfall forecasts can afford KRCS more flexibility and potential for early preparedness.

Having made these suggestions for the HRA we must acknowledge the importance of balancing detail with wide inter-pretability. In this case, although users such as KRCS may prefer to see more spatial detail in the advisories, in their current form the text-based county-level format means that no technical knowledge is required to correctly interpret the informa-tion. This facilitates understanding and easy dissemination (e.g. through radio, translation to local languages and in-person broadcasts to communities). To add additional information may limit the ease with which they are disseminated and their in-terpretability and accessibility. Ensuring an optimal balance for all stakeholders is a challenge for KMD and indeed for NMHS in general.

### 4.2.2 Improving flood forecasting

Explicit modelling of local hydrology is necessary to provide accurate forecasts of flood risk, rather than reliance on rainfall forecasts alone. Although here we do find that HRAs warn of the most significant flooding events (consistent with the analysis of Robbins and Titley (2018), who also find a good relationship between precipitation forecasts and heavy impacts across the globe), it is unlikely that flood impacts will always be felt after heavy rainfall. Or indeed it is not the case that heavy rainfall is always necessary to trigger flood impacts which can occur with 'normal' rainfall if the soil is already saturated (MacLeod et al., submitted). Accurate characterisation of flood impacts requires consideration of non-meteorological and non-hydrological factors.

A unified national flood modelling and forecasting system would provide KRCS with a standardized view of flood risk across the country, however KMD do not yet have such a system and different approaches are being followed in different basins. The Nzoia basin of western Kenya currently has the only operational flood forecast, where a basin-scale hydrological model is used to generate a three-day discharge forecast using basin-average rainfall and soil moisture observations along with a short-range rainfall forecast. Substantial new investment is being made in flood forecasting in Kenya, notably under the World Bank-supported Water Security and Climate Resilience project. This will both upgrade the Nzoia flood forecasting system with a new hydrological model software and will support an extension of river flood early warning systems to other main river basins of Kenya, including upgraded hydro-meteorological observation networks supporting hydrological flood forecast models. This will help to provide more targeted relevant flood forecasts, and as the hydrological monitoring network is expanded this will help to evaluate the background level of flood risk, supported by new hydrological model simulations. The work will also help to strengthen institutional links between KMD with the mandate for forecasting in Kenya and the Water Resource Authority (WRA) with the mandate for flood risk mapping; close collaboration between KMD and WRA is essential to ensuring effective and coherent flood risk management and forecasting in the region. Other parallel related activities include: the SHEAR HiPac project, which for the Nzoia river basin will map inundation risk in high resolution and link this to forecasts from the existing system, and the EU-supported ECHO project developing flood risk assessment and forecasting for the Tana River.

In the absence of readily available flood forecast information from the NMHS covering the entire country, some na-tional Red Cross societies are now considering the use of ECMWF GloFAS flood forecasts (see Alfieri et al., 2013, and

www.globalfloods.eu) to trigger flood EAPs. In Kenya GloFAS may be an appropriate product whilst the basin scale flood forecasting remains under development and there remains no unified national flood forecasting system. Whilst GloFAS is advantageous as it is freely available with national coverage, the GloFAS forecasts are unable to take advantage of real-time local hydrological observations to initialise the model, limiting the forecast skill. A locally-calibrated model which assimilates initial hydrological states would likely provide the optimal basin-scale flood risk forecast. In addition the need for GloFAS forecast verification remains outstanding for most basins. KRCS should work with relevant organisations to undertake this analysis. Further, use of GloFAS should be sensitive to issues of national ownership of warnings systems.

Ultimately the evaluation of HRA presented here should be put in the context of flood preparedness systems such as the KRCS flood hazard EAPs. It points to the need, now widely recognised, for strengthened co-production of forecast information and products which support the effective uptake of forecasts into risk management systems. In Kenya recent projects exemplify this approach including ForPAc, WISER SCIPEA and W2SIP, whilst the national Early Warning-Early Action platform convened by KRCS in September 2019 brought together relevant national actors. Co-ordinated verification of existing forecast products such as the HRA presented here will help to integrate these into systematic preparedness activities. Whilst in this case the current form of the HRA may preclude their use as a trigger for the KRCS EAPs, they are able to effectively warn of heavy rainfall and should therefore take a key role in a seamless approach toward mitigating the risk from risks associated with heavy rainfall across Kenya.

.

*Author contributions.* All authors collaborated on the development of the verification strategy and contributed to the manuscript. MK, EM and DM digitized the advisories, EM provided the KRCS flood record, MO prepared exposure data and PR extracted flood impact data from EM-DAT. DM co-ordinated the study, carried out the analysis and wrote the text.

*Competing interests.* The authors declare no competing interests.

*Acknowledgements.* The authors would like to thank two anonymous reviewers for their suggestions which led to strengthening of the analysis. In addition authors would like to thank KMD for provision of past heavy rainfall advisories and KRCS for sharing their record of flood events and estimations of flood-exposed population. This research was supported by the Science for Humanitarian Emergencies and Resilience (SHEAR) consortium project 'Towards Forecast-based Preparedness Action' (ForPAc, www.forpac.org), grant numbers NE/P000673/1, NE/P000568/1, NE/P000428/1 and NE/P000444/1. The SHEAR programme is funded by the UK Natural Environment Research Council, the Economic and Social Research Council and the UK Department for International Development. Further support for author M.A. came from the Innovative Approaches for Risk Protection (IARP) project funded by the IKEA foundation.

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

Tel: +2542038567880-5, +254724255153-4 Email: director@meteo.go.ke

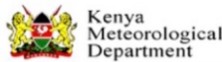 **Heavy Rain & Storm Advisory**

**Message Type:** Heavy rain /storm

**Message Update No.:** One

**Date of Origin:** 13th January 2016, 1200 UTC

**Validity:** 15th to 17th January 2016

**Severity:** Mild to Moderate

**Certainty:** Probability of occurrence (33%)

**Message Description:** Rainfall of more than 30 millimeters is likely to occur over some areas of Mount Kenya and South Rift Regions on 15th and 16th, including Nairobi and Kiambu on 17th January 2016.

**Area(s) of Concern:** These areas include Narok, Bomet, Kericho, Nakuru, Nyahururu, Nyeri, Muranga, Embu, Meru, Kiambu and Nairobi.

**Instructions:** Residents in these areas are advised to be on the lookout for sudden downpours which may cause flash floods. They are advised to exercise caution especially if these rains persist for a long time in one place. Further advisories will be issued as we follow up on the progress of this weather event.

**Message Addressed to:** Media, County Directors of Meteorological services in affected areas and other emergency response institutions.

**Originator:** Director, Kenya Meteorological Department-Headquarters Nairobi

**Figure 1.** An example of a heavy rainfall advisory issued by KMD.

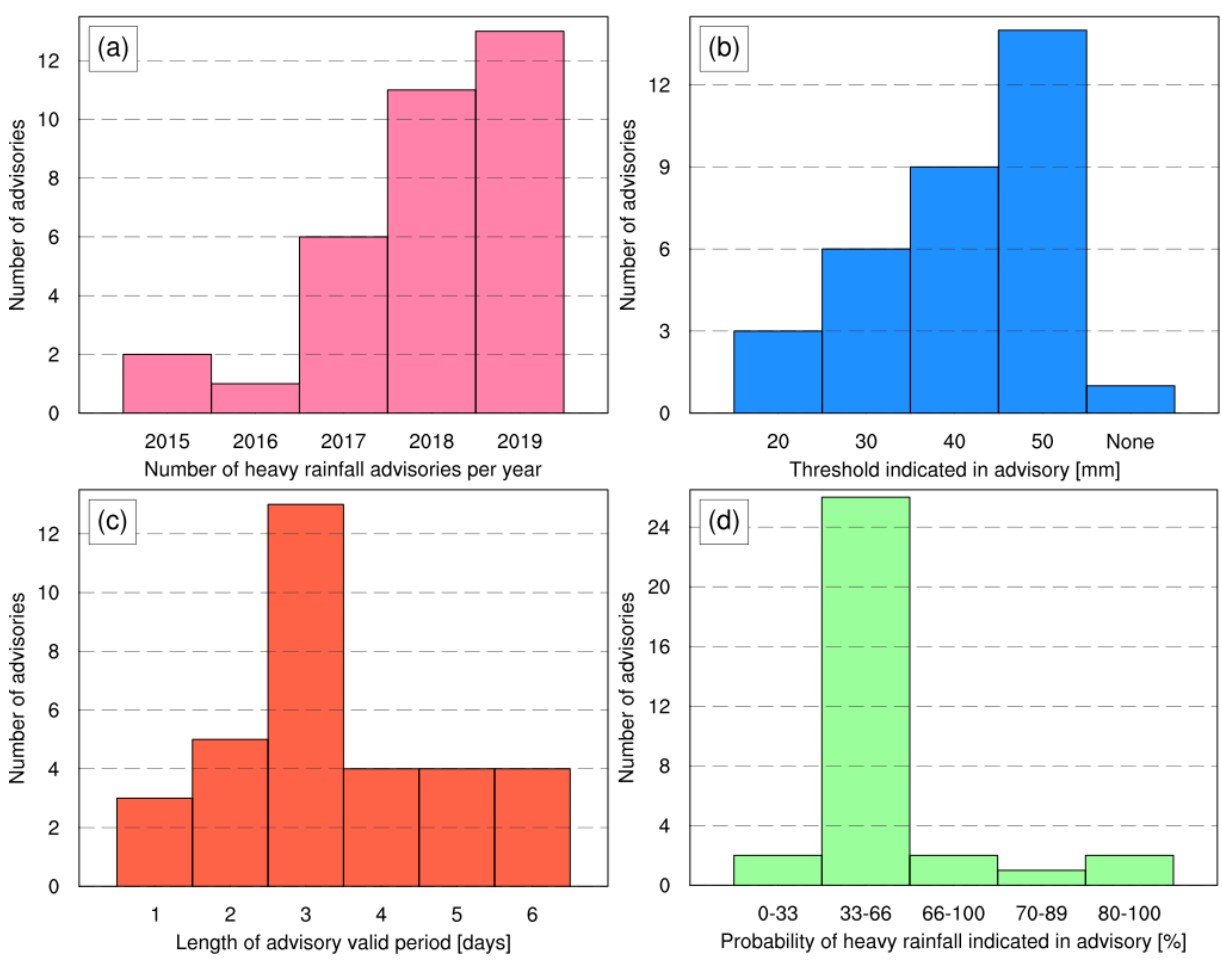

**Figure 2.** Summary statistics of advisories issued over 2015-2019 detailed in Table 1. Showing (a) the number of advisories issued per year, (b) the rainfall threshold mentioned, (c) the length of the valid period and (d) the probability mentioned.

## (a) Counties warned in each advisory

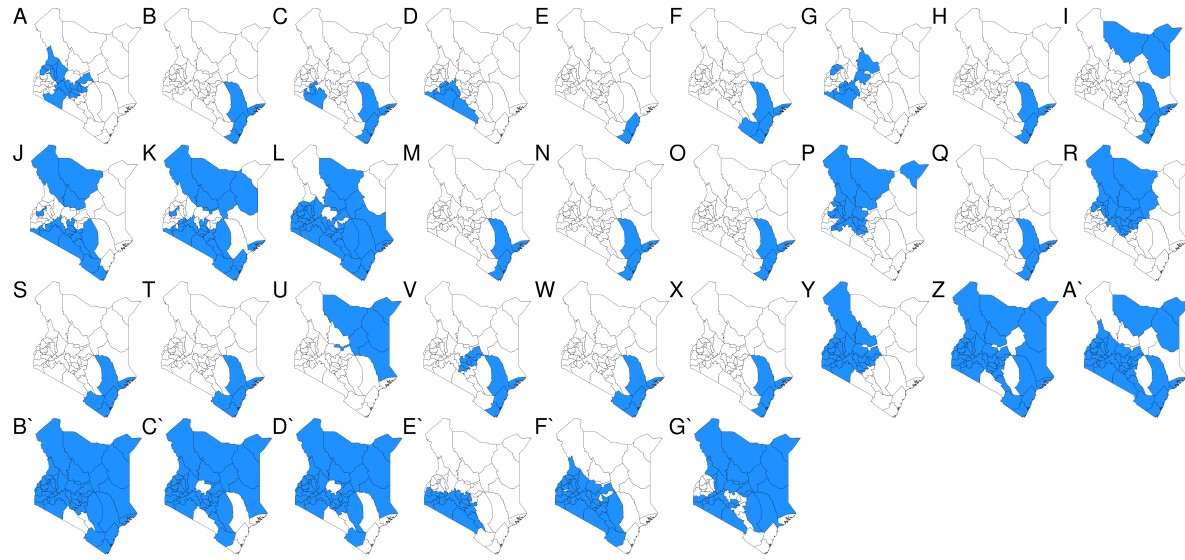

## (b) Accumulated rainfall during each advisory window

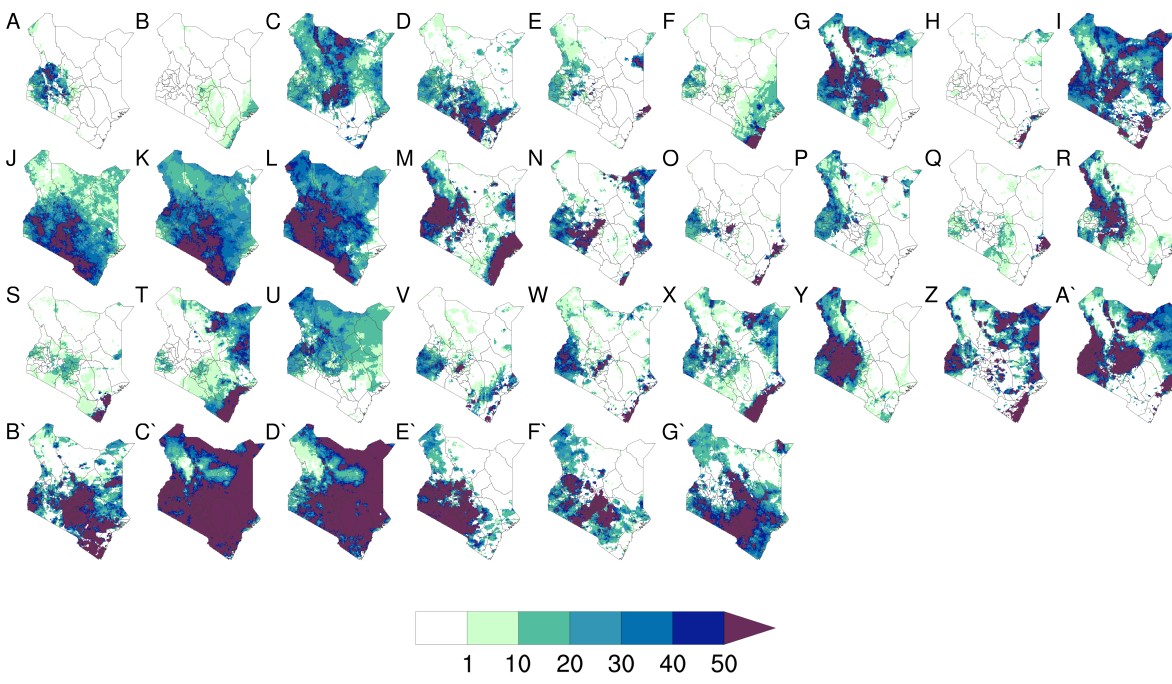

**Figure 3.** (a) Counties with active warnings for each of the 33 heavy rainfall advisories issued by KMD during 2015-2019 (advisory details are given in table 1). (b) Rainfall accumulations (mm) during each advisory window, based on CHIRPS.

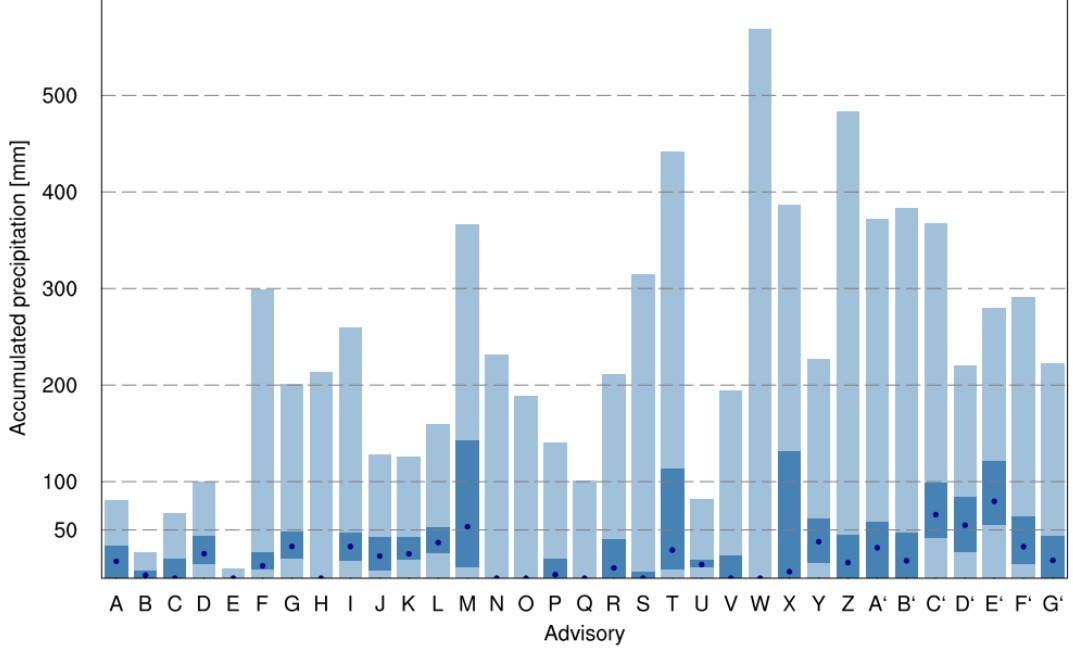

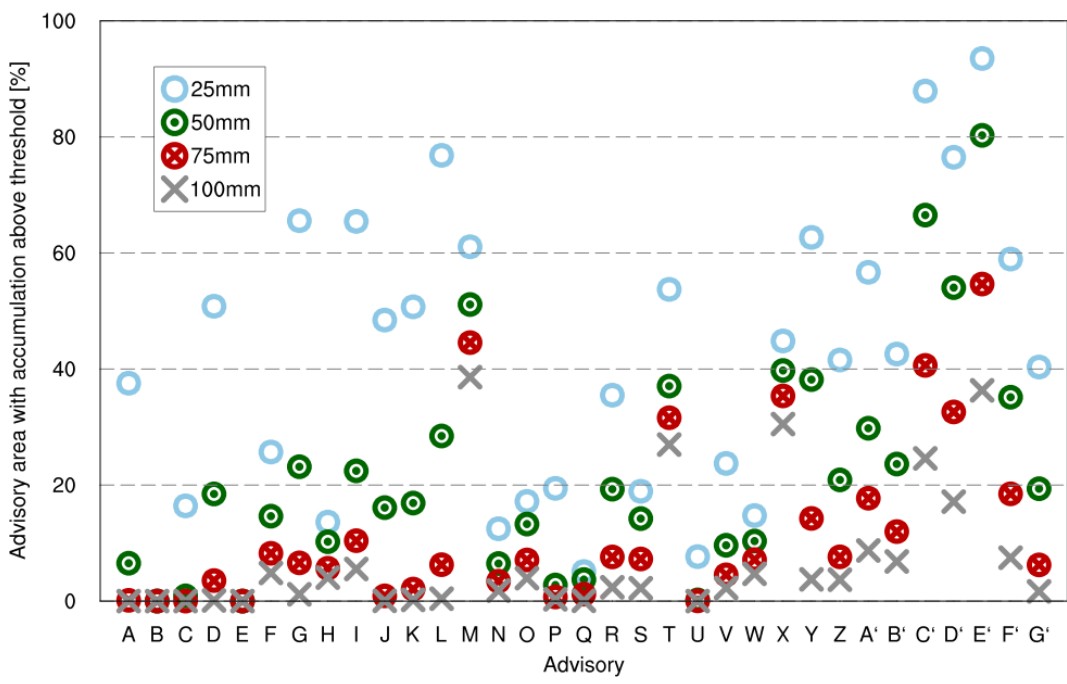

**Figure 4.** How much rain fell in counties under advisory? (a) Rainfall accumulation during advisory window, showing distribution over all 5km square gridpoints within counties mentioned in advisory (light/dark shading shows the range/interquartile range of the distribution, and the dot indicates the median). (b) Percentage of each advisory region where rainfall accumulation was above 25, 50, 75 or 100mm.

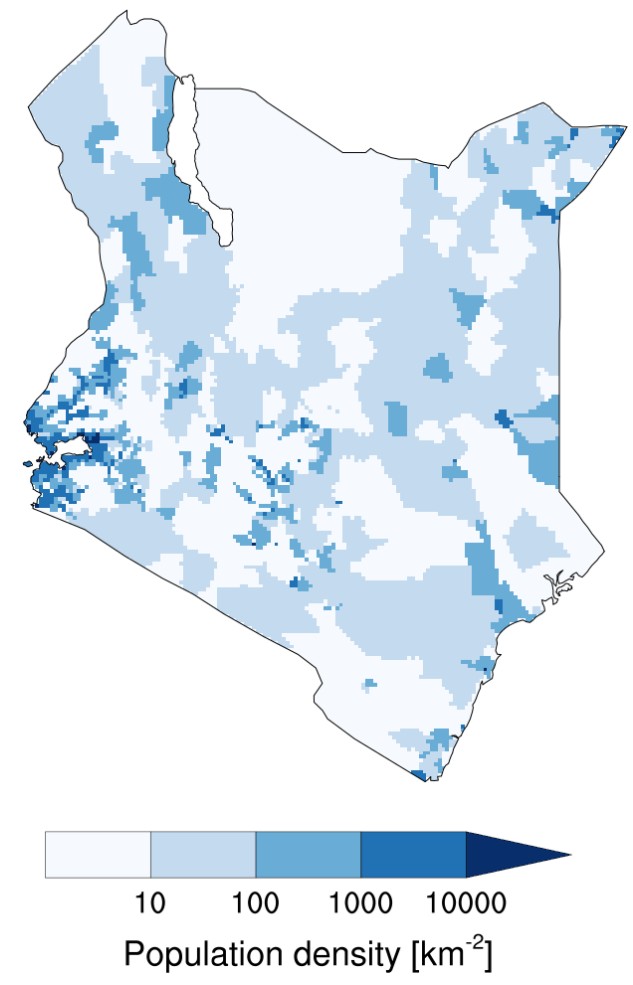

**Figure 5.** Population density over Kenya, from the Gridded Population of the World Database produced by NASA SEDAC (CIESIN, 2018)

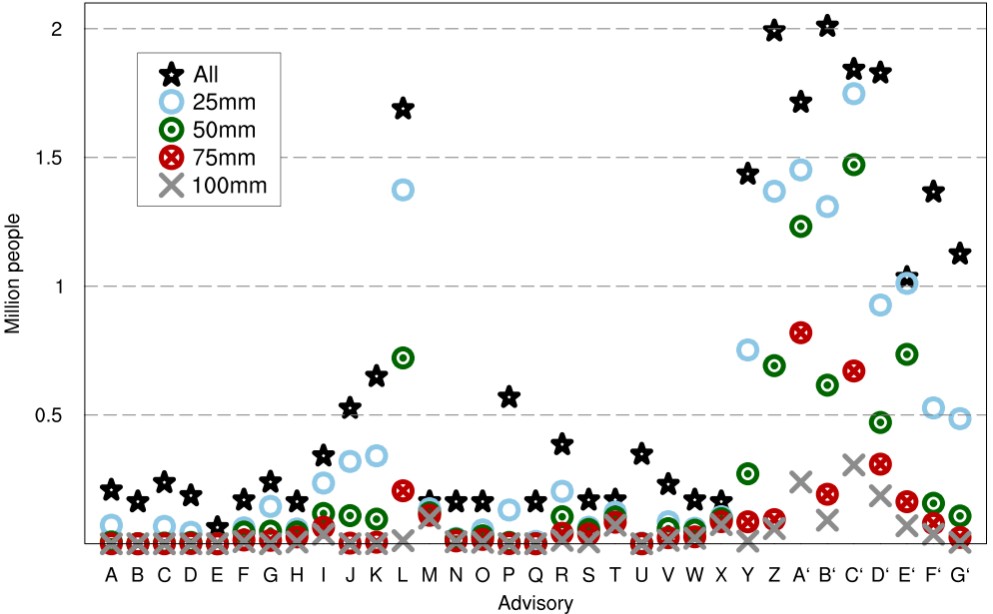

(a) Exposed population under advisory receiving threshold accumulation

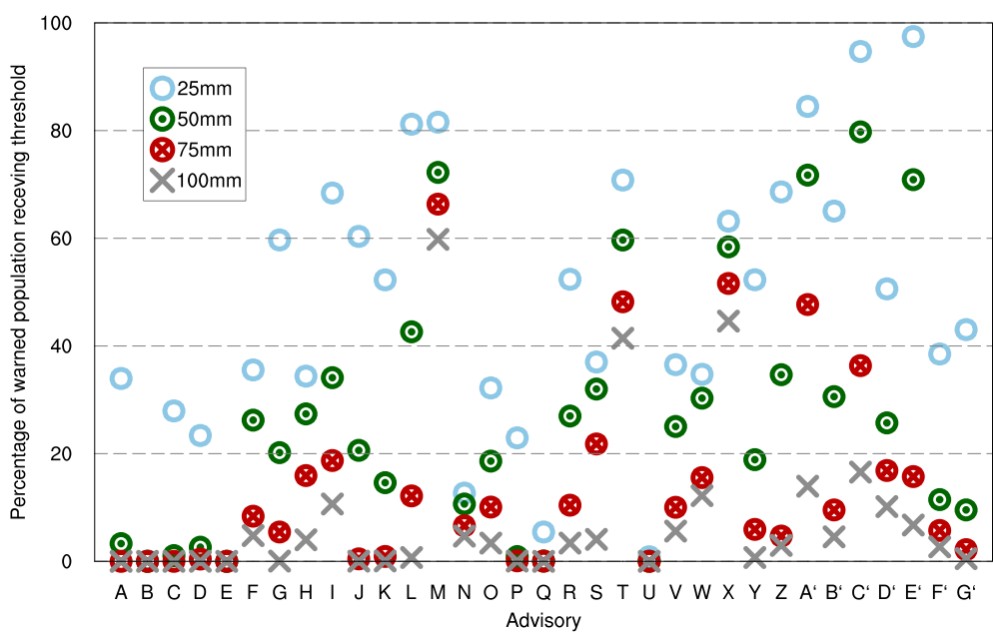

(b) Percentage of exposed and warned population receiving threshold accumulation

**Figure 6.** What is the extent of preparedness action implied by advisories? (a) The total population living in the warning region (black star) and the number living in that region also receiving at least 25, 50, 75 or 100mm rainfall over the advisory window. (b) Percentage of the population living in the advisory region and also receiving above-threshold rainfall.

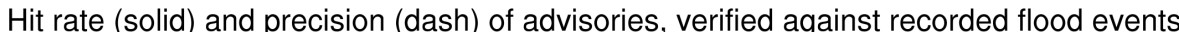

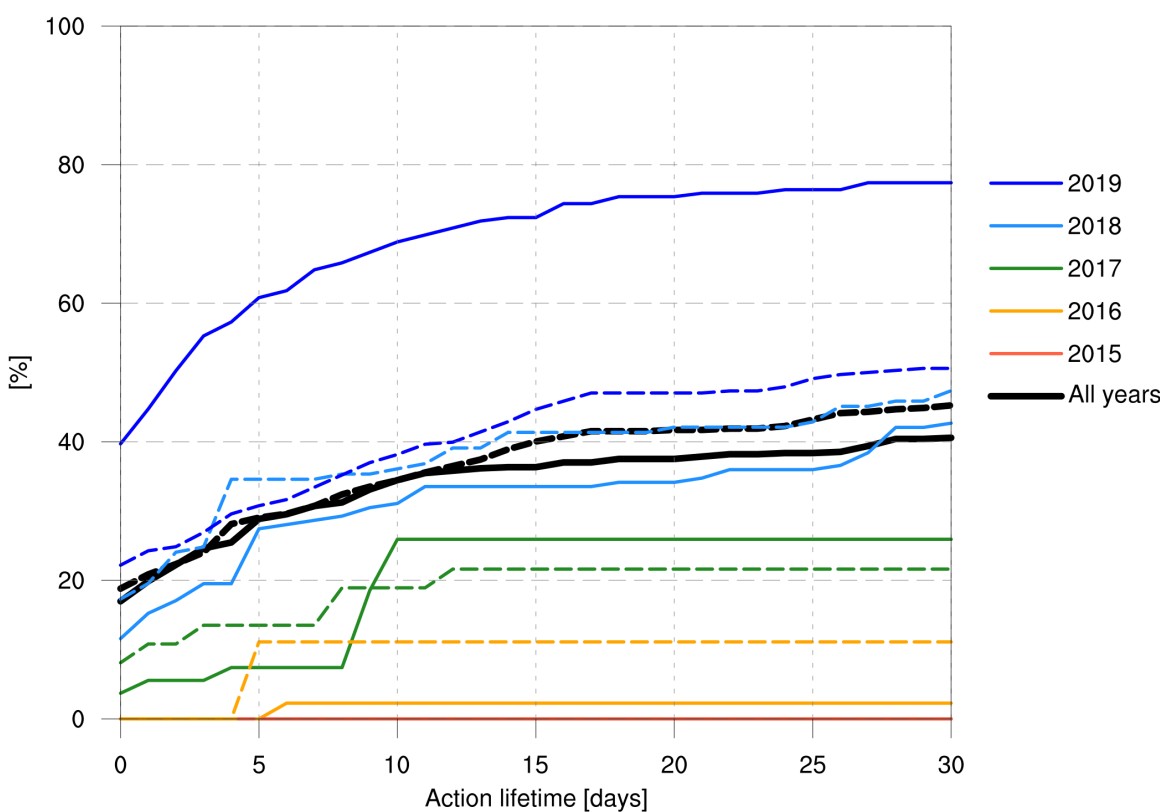

**Figure 7.** Skill statistics of the advisories when verified against observed flood events at county level. The hit rate shows the percentage of events which were preceded by an advisory in that county (solid line), whilst the precision shows the percentage of county warnings which were followed by an event (dashed line; NB, precision is equivalent to 100% minus the false alarm ratio). Statistics are calculated for all years (black line) and each year separately (coloured lines), across a range of 'action lifetimes', such that theoretical action based on each advisory is assumed to have a lifetime, so is still considered a 'hit' as long as the flood event occurs within the lifetime of the action.

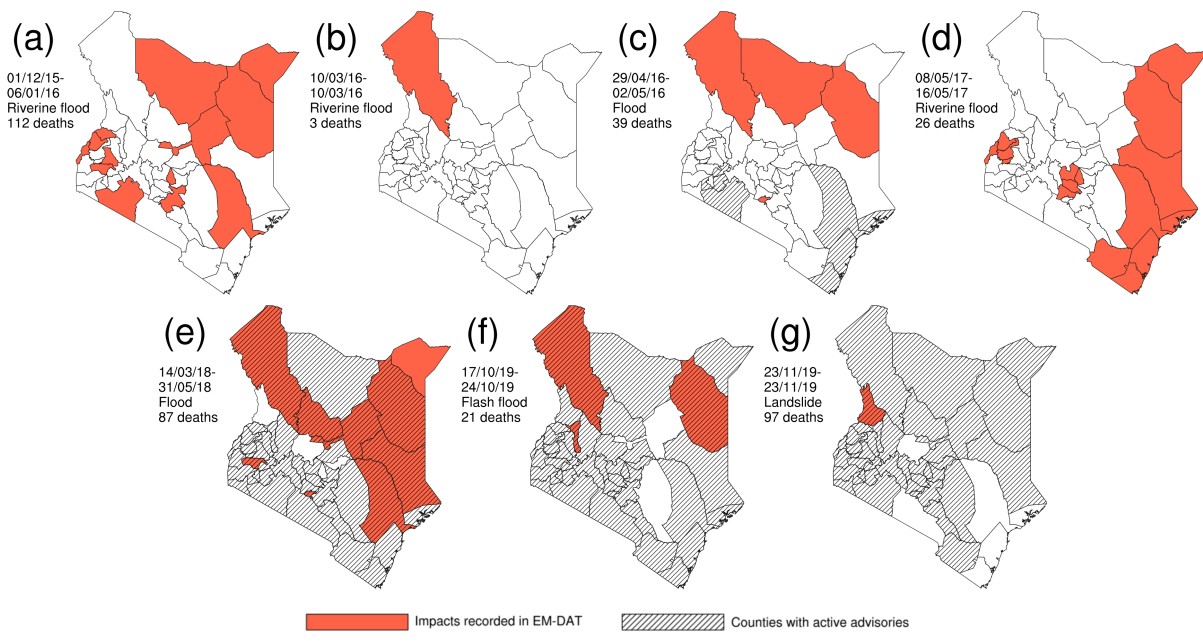

**Figure 8.** Were the most significant impacts of heavy rainfall preceded by advisories? Showing all seven relevant events extracted from EM-DAT across the advisory period (see section 2.3 for details of event selection). Counties reporting impacts are shown in orange, whilst hatching indicates counties for which warnings were active when the impact was recorded to have begun.

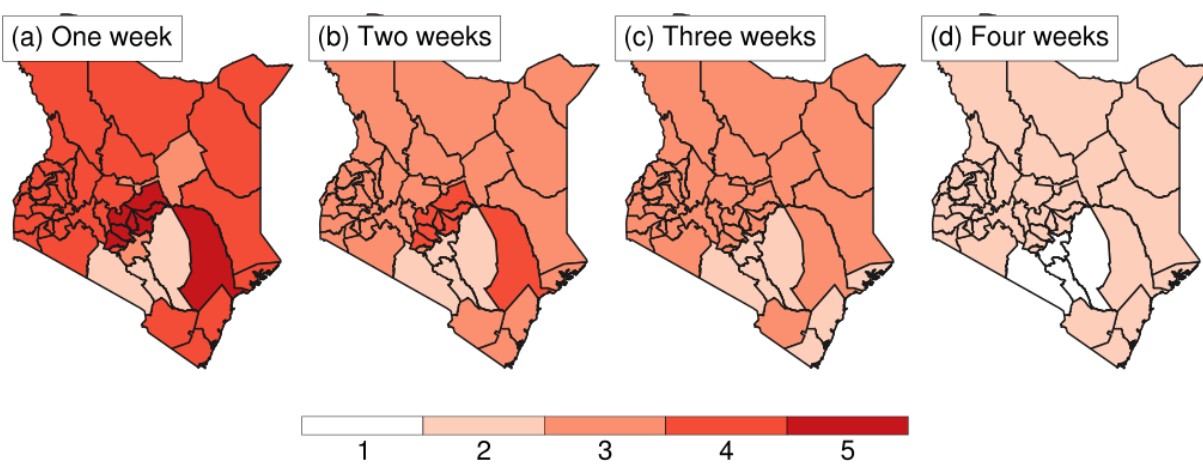

**Figure 9.** How many times per year might an FbF/A system based on advisories trigger? Showing the number of potential triggers per county during 2019: here we assume that an action is triggered if an advisory is issued, as long as no action had already been triggered in the preceding one, two, three or four weeks (a-d).

**Table 1.** Summary of all advisories 2015-2019 evaluated in this study

| Label | Issue date | Period length (days) | Largest rainfall threshold mentioned | Probability indicated |
|---|---|---|---|---|
| A | 2nd June 2015 | 2 | 50mm | 33-66% |
| B | 2nd July 2015 | 2 | 50mm | 0-33% |
| C | 25th April 2016 | 2 | 50mm | 80-100% |
| D | 18th April 2017 | 2 | 50mm | 33-66% |
| E | 28th April 2017 | 1 | 50mm | 70-89% |
| F | 18th September 2017 | 3 | 50mm | 80-100% |
| G | 11th October 2017 | 3 | 50mm | 33-66% |
| H | 30th October 2017 | 2 | 50mm | 33-66% |
| I | 2nd November 2017 | 4 | 30mm | 66-100% |
| J | 27th February 2018 | 3 | 50mm | 33-66% |
| K | 9th March 2018 | 4 | 40mm | 0-33% |
| L | 15th March 2018 | 4 | 50mm | 66-100% |
| M | 27th April 2018 | 5 | 40mm | 33-66% |
| N | 2nd May 2018 | 3 | 50mm | 33-66% |
| O | 7th May 2018 | 3 | 50mm | 33-66% |
| P | 20th May 2018 | 1 | 50mm | 33-66% |
| Q | 30th May 2018 | 1 | 30mm | 33-66% |
| R | 4th June 2018 | 3 | 40mm | 33-66% |
| S | 24th September 2018 | 3 | 50mm | 33-66% |
| T | 23rd October 2018 | 3 | 40mm | 33-66% |
| U | 25th March 2019 | 3 | 30mm | 33-66% |
| V | 3rd May 2019 | 4 | 40mm | 33-66% |
| W | 7th May 2019 | 5 | 30mm | 33-66% |
| X | 22nd May 2019 | 3 | 40mm | 33-66% |
| Y | 31st May 2019 | 6 | 40mm | 33-66% |
| Z | 10th October 2019 | 5 | 20mm | 33-66% |
| A' | 14th October 2019 | 5 | 40mm | 33-66% |
| B' | 23rd October 2019 | 6 | 20mm | 33-66% |
| C' | 18th November 2019 | 6 | 40mm | 33-66% |
| D' | 23rd November 2019 | 3 | 30mm | 33-66% |
| E' | 28th November 2019 | 6 | 30mm | 33-66% |
| F' | 3rd December 2019 | 3 | None | 33-66% |
| G' | 6th December 2019 | 3 | 20mm | 33-66% |