# Peer review of "Are Kenya Meteorological Department heavy rainfall advisories useful for forecast-based early action and early preparedness for flooding?"

_Natural Hazards and Earth System Sciences, 2020_

## Referee Comment (RC1) · Anonymous Referee #1 · 1 Jul 2020

Referee comment

Review of "Are Kenya Meteorological Department heavy rainfall advisories useful for forecast-based early action and early preparedness?"

Authors: David MacLeod et al.

MS No.: nhess-2020-122
* * *
General comments

[Figure]

This manuscript provides an evaluation of 33 Heavy Rainfall Advisories (HRA) issued by the Kenya Meteorological Department (KMD) during 2015-2019. This analysis is potentially useful for forecasters, practitioners, and decision makers concerned with the prediction of natural hazards and communication of warnings for early action in the region. Since it is essential to evaluate the skill of operational early warning systems, the first assessment of these advisory warnings for Kenya reported in this manuscript is of great practical value for the community. Such an analysis has the potential to provide some evidence-based recommendations for future improvements of similar heavy rainfall advisories in Kenya and in similar contexts.

In general, the article is quite well written and easy to read, even if some specific parts should be improved by making the text clearer, providing some more context and motivations, giving some important details or references to support some statements/assumptions (see major and specific comments below).

However, the manuscript needs some major revisions: the authors should make more efforts in terms of analysis to address the questions posed here more thoroughly, improving some methods to provide more quantitative elements on whether these HRAs are useful and how they could be improved, but also discussing more thoroughly the current barriers and limitations of the HRAs (especially the spatial detail issue, see comments below) and how these fit within the context of current and future co-production efforts. Some justifications used to support a qualitative (or proxy-based) analysis are not convincing. The proxy/qualitative indicators that are used to answer two central questions in the article (questions 1 and 2, see page 5) can provide only partial insights and non-robust indications on the usefulness of the advisories (given unrealistic or not convincing assumptions on relevant trigger probabilities and population exposed to flooding). So far, some parts of your analysis cannot convincingly support a few central points of your conclusions. Thus, major revisions are recommended.

———————————————

[Figure]

Major comments

1. The first major concern is that to estimate the relative scale of preparedness implied by each advisory, the population exposed and vulnerable to heavy rainfall and consequent impacts (flash floods, water-logging or riverine floods) should be used instead of the total population for each warned county. The total population living in a warned area seems an oversimplified and unrealistic proxy indicator, that does not provide a measure of the number of people likely to benefit from flood preparedness actions in the region and does not allow a comparison of the extent of preparedness action required between advisories. The authors partly recognize this issue, but do not address it properly and do not convince the reader on the value of their 'first-guess' estimates. A proxy estimate based on the total population per county does not seem a sensible approach even to provide a broad indication of the relative amount of preparedness appropriate for each advisory (see L. 170-174). The broad indication derived from this proxy could deliver the wrong message (or maybe the right message but for the wrong reasons), being based on assumptions that are not necessarily true (i.e. there is some correlation between total population of a county and vulnerable/exposed population to heavy rainfall per county, but the distribution of vulnerable population across counties may not match the distribution of total population).

Related to this, it should be also acknowledged that although there is an attempt at overlaying population density and rainfall accumulation observed over each advisory window (L. 262-265 and Figure 6b), in many cases the population living in an area receiving heavy rainfall does not coincide with the population potentially affected by flooding, especially for riverine flooding events. Please consider using some additional datasets of population potentially affected by flooding. For example, you might want to use some datasets that exist to better estimate potentially affected population by flooding at least in some areas, based on data available for past events in some regions of Kenya, at least to provide a case-study example of the extent of preparedness implied by a single or few advisories, e.g. for eastern Kenya you could use the

dataset available in the OCHA's Humanitarian Data Exchange (HDX) platform, based on Sentinel-1 imagery acquired on 2018 : https://data.humdata.org/dataset/potentially-affected-population-by-flooding-in-eastern-kenya-2804

2. A second major concern is that triggers should be defined more realistically, based on some relevant rainfall thresholds and effective probability triggers. You argue that you may avoid considering any specific rainfall threshold or probability because it would not 'provide robust statistics and precludes any meaningful statement' (e.g. see lines 138-140). However, while I agree that the sample size and the inconsistencies of the data from these 33 HRAs preclude any meaningful calculation of some verification metrics such as the reliability of probabilities, I think that the available probabilities and rainfall data could still be used to answer the questions in your paper more quantitatively. In other words, I agree that you cannot compare warnings with different levels of spatial aggregation, different temporal windows for accumulation of rainfall, etc., but you can still test whether the HRAs were useful overall for forecast-based early action based on some minimum quantitative analysis. For example, you can set a minimum rainfall threshold (maybe dependent on the window of accumulation, or a minimum with a larger window) and some significant probability based on the classes available, e.g. probability of heavy rainfall > 33% (and not just above zero). The main problem I see in your analysis is that you have defined the action trigger as the probability (of heavy rainfall) exceeding zero, but an action trigger with a very low probability of unspecified heavy rainfall level seems a very unrealistic trigger even for low-regret actions. Such an approach is likely to lead to overconfident verification results on the value of the HRAs. For example, would a probability lower than 10% of heavy rainfall expected to fall over a big county still lead to any concrete action by government or humanitarian agencies? If you have valid reasons to think so and use a very low trigger probability, you should at least extend the discussion on this point to convince the reader of the validity of this analysis. Maybe you could support this choice based on some literature, or any reported practice in the humanitarian sector, explaining how this would be useful and what actions would be informed – otherwise all the analysis and conclusions seem

to be based on unrealistic assumptions. Still, I believe that the analysis would be more valuable if you could show the performance of the warnings for significant probability levels and considering some meaningful rainfall thresholds that are more likely to be used as triggers.

If you defined triggers in a specific and realistic way, this would allow to make your analysis more concrete and link it to some specific actions to determine the extent to which the KMD HRAs could guide 'worthy' preparedness activity. Your definition of 'worthy' action seems being kept purposely vague (in line with a zero-probability trigger) and not clearly defined, referring to any preparedness assistance and no particular action (e.g. line 181). Using some specific examples of actions and trying to quantify whether taking these actions would be worth would be a natural step forward to give more concrete value to your analysis. Please consider including some specific action-based analysis or examples that could give more value to the article.

3. There is a lack of evaluation of misses (missed events in the warnings) or at least a discussion on it: the evaluation of observed events is only based on seven reported impactful events (the most significant floods events in the EM-DAT dataset), and there is no proper evaluation of 'misses' and 'hit rates' based on a large sample, which is a limitation of the data and period available. Despite the obvious sample size issues, it would be probably possible to include in the analysis in Section 3.1/Section 3.3 some more information on observed events also based on other data (not only EM-DAT).

Are there any other significant flood events beyond the 7 events from EM-DAT (e.g. maybe events with less than 10 fatalities but still high number of affected people / households affected or damaged) that have been missed by the HRAs during the study period? Please discuss this.

If more events were available (beyond EM-DAT), in Section 3.1 you could include an evaluation of hit rates per county using CHIRPS as reference and/or in Section 3.3 a proper evaluation of misses based on the larger sample of reported impactful events.

Without a full evaluation of hits and misses, the 'hits' picture that is given might be misleading and uncomplete. You mentioned some misses because related to the time windows of advisories issued (e.g. for advisories warning "wrong" counties, see lines 233-234). It would be useful for decision makers if you could calculate a proper hit-rate even if based on a sample of 33 advisories. You could focus on a specific trigger probability and threshold rainfall, for example you could keep the 50 mm nominal threshold case and use a specific probability threshold. Of course, using only a single rainfall threshold across a big country as Kenya is not a proper location-specific indicator of flood impacts, but this would be still useful. Additional analysis with some more observed events (if you had more than these 7) would probably help understand also whether the step change in the advisories in 2017 (access to GHM) reduced the number of misses, as it seems the case from your analysis based on 7 events (Figure 7) and might be expected from the increasing number of advisories per year (Fig. 2a). Section 3.3 could then be more complete by focusing on both hits and misses, by reporting how many observed events were not preceded by advisories.

4. The analysis provides useful insights on the possible limitations of the HRAs and reccomendations, but the discussion should make more efforts in understanding and explaining the current limitations of the HRAs. This is essential to provide more specific recommendations for improvement of the HRAs. One of the major limitations of the HRAs that arises from the analysis is the lack of precise spatial information in the warnings beyond the county-level information. As you suggest, free-shape warning areas should be used instead of administrative county boundaries. However, you also explain that such free-shape warning polygons are currently generated by the GHM and are already in use at KMD. Thus, "KRCS could then overlay these with maps of population exposure and vulnerability to flood risk, in order to further narrow down targets for intervention". Then why more precise warnings are not issued yet?

It is not clear what is the current bottleneck for providing more spatial detail in the advisories, and it would be important to understand whether there are either scientific

/ technical or institutional / economical barriers that prevent this, e.g. either whether it's a lack of resources (e.g. GIS technicians) at KMD, or whether there has not been enough co-production effort made so far to enable a full use of the GHM information at KMD, or whether there are some information layers missing (e.g. flood extent maps which do not coincide with heavy rainfall extent). Without an extensive discussion on this point, it is not clear how the spatial detail in the advisories could be improved.

5. Finally, it would be essential to discuss in this article how the HRAs fit within the need "for strengthened coproduction of forecast information and products" (now widely recognised as you also recognise). Is there any issue of national ownership in the use of GHMs from the UK MetOffice in the HRAs?

From your article, it seems that only the subjective interpretation of the forecasts and the writing of the HRAs summary is carried out within KMD and so within the mandated agency in Kenya. How is this perceived at the national level? I can see that the actual sources of forecast information in the HRAs are not mentioned in the HRAs (see Figure 1, no field on 'data sources') so maybe there is no general perception from the communities involved in Kenya or even from county directors / national policy makers around that issue. However, this point would need attention in such a paper. Also, it would be useful to detail whether KMD get access to raw forecast data from ECMWF and UK MetOffice or only to some end-product maps and the level of spatial detail in these maps (e.g. do KMD get any shapefiles or netcdfs data? At which resolution?). This point might help explain one of the major current limitations (lack of spatial detail in the HRAs) and how KMD and their international partners could deal with it to improve the advisories.
* * *
Specific comments

- L. 34: it would be good to specify how this UK-funded project fits within the local context and is linked with other projects mentioned (IARP) or not; are these projects

making some efforts in coproduction and capacity building and how specifically (only by giving the outputs of GHMs models to local agencies or is there something more, e.g. capacity building efforts)? It would sound very sensible to discuss this.

- L. 45: do KMD work on their own on hydro-meteorological forecasting? The mandates and institutions involved in hydrological warnings in Kenya should be clarified, as KMD is the meteorological agency, and there is also a national hydrological agency, the Water Resources Authority (WRA, https://wra.go.ke/) that should be responsible of flood forecasting activities alongside KMD (e.g. see FLOOD ADVISORIES issued by WRA; see also ODI working paper 553, April 2019, "Reducing flood impacts through forecast-based action" by Lena Weingärtner et al.). The links between KMD and WRA are not clear nor mentioned in the paper. It seems an important point to discuss, as probably a closer collaboration with forecasters at WRA could help make the HRAs by KMD more precise, with impact-based focus and hydrologically meaningful. Is there a link between the KMD Flood Forecasting Unit and WRA? Or is this a current institutional barrier? Hydrological forecasting is at the interface between met and hydro agencies not only in Kenya but in many countries and similar questions may arise elsewhere. Some more context about this important point should be provided.

- L. 101: I would suggest specifying "The GHM system", to avoid confusion with other possible meanings, e.g. HRA or other systems just mentioned in the text

- Section 2.2 (Verification Approach): L. 150-219 are difficult to follow and should be reorganised in a more clear or compact way (e.g. with bullet points for all the methods and data associated to each question).

- L. 200-206: the part about the dam collapse of May 2018 seems excessively long in the context of this section and should be kept more concise; as it stands, that part does not flow well within the paper.

- L. 243-244: "As these advisories associate each warning with a probability, these findings are quite consistent" – this sentence is obvious and not specific enough. You

could try to add something more relevant and specific, as the previous remarks in terms of convective rainfall and percentage of warned area are interesting. For example, could you see visually any link between percentages of area warned which receive rainfall accumulation above the 50mm threshold and geographical locations/counties that are known to be more subject to convective rainfall?

- L. 347: "These kinds of actions would have significant costs, so more than ten triggers in a year may not be realistic" – that sounds reasonable but too approximate to be stated in this way, could you provide more details (e.g. approximate estimate of costs and resources) and improve the sentence? Are there any references supporting this sentence (and the number of ten triggers)? Are there any estimates in the scientific/economic literature or in reports of humanitarian agencies on the resources that would be needed / are available for early action and flood preparedness in Kenya or maybe in any specific region within the country?

- L. 405: "For this purpose they are effective" sounds a bit overstated, e.g. given the lack of spatial precision that you highlighted in the paper, the large area warned by identifying counties in the text may not be effective (people in an affected county may not take county-scale warnings so seriously, if these are preceded by warnings that were not followed by any event in their specific area in that warned county).

- "Data and verification approach" section: some final parts do not flow well and could be improved (e.g. more clear organisation by points and questions addressed); some parts need more details or references to the scientific literature or humanitarian reports to back-up some assumptions made (e.g. see also remarks in major points above).

- "Discussion and recommendations" section: there is a lack of discussion on the temporal consistency in the HRA dataset. There is a difference in the source rainfall forecast in the new data in recent years, but also the number of HRA has increased. So, what role does this inconsistency play in comparing earlier years with more recent ones? For example, the number of hits in recent years is expected to be higher simply

because there are more warnings issued.

————————————————————————

Technical corrections

- Table 1, column 3 - "Period length" is missing the units (days, probably)? - L. 2 (Abstract): "Forecast-based Action/Finance (FbA)", it seems that Forecast-based Financing is more used than Finance, please check. - L. 19: Climate risk or better hydro-meteorological risk? - L. 27: repetition to avoid in (see Wilkinson, for...) as L. 23 already mentioned it - L. 29: 'individual expenses' and/or probably even more 'community expenses'? - L. 164-165: it's fine to focus the discussion on results for 50mm accumulation, but maybe you can say here more explicitly that you took this threshold as "working definition for heavy rainfall", as sometimes later in the results section this is the wording you use (e.g. Line 246), so good to define it clearly from the methods, still mentioning the limitations of it as you do. - L. 167: "To answer question .. we estimate" is missing the question number - L. 399-400: see sentence "We find that an increase in skill over time, and that...", to be corrected. - L. 438: higher-cost actions - L. 456: repetition, "would would" - L. 470-471: repetition of "in Kenya" - L. 484: "mitigating the risk from risks", I would avoid the repetition - Figure 4a, caption: please clarify whether by "inner and outer quartiles" you mean "inner and outer fences" (which seems more common wording in this context)? – see "(dark/light shading shows inner/outer quartiles and dot indicates the median". (by the way, a parenthesis is missing there).

---

## Referee Comment (RC2) · Anonymous Referee #2 · 20 Jul 2020

GENERAL COMMENTS: This short paper gives an insight into how flood warnings are generated at the Kenyan weather service and how their skill evolved over the last 5 years. Despite the relatively small number of cases and some data inhomogeneity, I find the paper useful for practitioners and generally welcome publication of such work. Overall the paper is well written and logically structured. There is, however, substantial room for improvement with respect to data and the evaluation methodology as detailed in the following major and minor comments.

MAJOR COMMENTS:

1.) Evaluation procedure: Classically one would consider hits, false alarms, missed

events and correct non-events. This would enable the computation of all the classical scores such as Proportion Correct, Heidke Skill Score etc. Your analysis gives a good idea of hits and false alarms but the missed events are only treated with respect to the 7 flood cases from the EM-DAT database. Can you not use CHIRPS to give some idea for missed heavy precip events that you could define to have a certain intensity and spatial reach (as pointed out in Point 2 of Reviewer 1)? After that, all days that remain would be correct negatives. This would allow a more quantitative treatment of skill.

2.) Language: Overall the paper is nicely written and the level of language high. However, some passages are a bit wordy and redundant and I would therefore ask the authors to careful assess the potential for shortening. Given your overall low levels of statistical significance, I would also be a little more cautious with statements on skill throughout the text.

3.) Abstract: In its current state the abstract does not really explain well what the paper is all about and in what way it is important, new and special. There should be more information on data, method, results and limitations.

4.) Rainfall data: This is always an issue. There are many different products with strengths and weaknesses. Please provide more evidence that CHIRPS is a good one (the best?) to use and possibly repeat exercise with an alternative source of information.

5.) Section 2.2: I think that the approach you are taking is largely well conceived (but note my reservations under Point 1) given all the restrictions at hand but the section as written is quite long and your quantitative metrics are only described and nowhere cast into formulas. I suggest giving this section a clearer structure and a more "recipe like" description of how you compute metrics. If you give names or abbreviations to your metrics, you would not need to repeat the description again in Section 3.

6.) EM-DAT: I find the thresholds of 10 deaths too high and would feel that even one death would justify a weather warning. Given that you have authors from Kenya that

may have access to government documents, is there no alternative source of information that would give you a list of flood events of smaller magnitude, too? This would much improve your statistics relative to the few events in EM-DAT!!

7.) Population numbers: I agree with Reviewer 1 that a distinction between all population of a county and the fraction likely affected by floods (in particular riverine) would be desirable. However, I can imagine that such fractions are not easily available and feel that the paper would be of value without it. In this case the authors could raise this point more clearly in the text and give at least some orders of magnitude from literature.

MINOR COMMENTS:

1.) Punctuation: There are a lot of places with inconsistent or suboptimal use of commas. Please check carefully throughout the entire manuscript.

2.) L2: remove "a" as in plural

3.) L5-6: What are you trying to say with this sentence. Please reword!

4.) L12: no comma

5.) L19: is it really a "movement"? In L31 it is called a "society"?

6.) L30: IFRC?

7.) Section 1: this gives a nice introduction to the topic but some bits are a little redundant and could be streamlined.

8.) L75-76: avoid repetition of "improve"

9.) L120: remove period after figure 2

10.) L125: better turn this into a proper sentence

11.) L167: this question?

12.) L194: requires?

13.) Table 1: Why don't you merge the first two entrances?

14.) L234: fell during . . .

15.) L241: "quite a reasonable chance" is very fuzzy, reword!

16.) L245-248: What result or figure does this paragraph refer to?

17.) Figure 4 could be discussed in a little more detail.

18.) Figure 5 I would rather include in the Methods section 2. You can then also discuss there the difference between all people and those affected by a given flood (see above).

19.) L255: remove "extreme" as upper bound is already an extreme

20.) L286: highest number?

21.) L319: on 18th November?

22.) L385: I would maybe not use the word "all" here, as it remains a probabilistic problem, where some missed events are unavoidable.

23.) L441: double period

24.) L443: comma instead of period

25.) L456: 2x would

26.) L458-59: not a proper sentence

27.) Section 4.2.2: Too much detail to my taste. This is a scientific paper and not a government technical document.

28.) Figure 2 caption: include that these statistics are done for the cases listed in Table 2.

29.) Figure 4 caption: these should be 5kmx5km gridpoints

30.) Figure 5 caption: two brackets at end

31.) Figs.: I would generally not start a caption with a question.

---

## Author Comment (AC1) · 26 Aug 2020

**NHESS-2020-122: Are Kenya Meteorological Department heavy rainfall advisories useful for forecast-based early action and early preparedness?**

**Response to review RC1**

Original comments are duplicated below in blue, with our responses following in turn.

This manuscript provides an evaluation of 33 Heavy Rainfall Advisories (HRA) issued by the Kenya Meteorological Department (KMD) during 2015-2019. This analysis is potentially useful for forecasters, practitioners, and decision makers concerned with the prediction of natural hazards and communication of warnings for early action in the region. Since it is essential to evaluate the skill of operational early warning systems, the first assessment of these advisory warnings for Kenya reported in this manuscript is of great practical value for the community. Such an analysis has the potential to provide some evidence-based recommendations for future improvements of similar heavy rainfall advisories in Kenya and in similar contexts.

In general, the article is quite well written and easy to read, even if some specific parts should be improved by making the text clearer, providing some more context and motivations, giving some important details or references to support some statements/assumptions (see major and specific comments below).

However, the manuscript needs some major revisions: the authors should make more efforts in terms of analysis to address the questions posed here more thoroughly, improving some methods to provide more quantitative elements on whether these HRAs are useful and how they could be improved, but also discussing more thoroughly the current barriers and limitations of the HRAs (especially the spatial detail issue, see comments below) and how these fit within the context of current and future coproduction efforts. Some justifications used to support a qualitative (or proxy-based) analysis are not convincing. The proxy/qualitative indicators that are used to answer two central questions in the article (questions 1 and 2, see page 5) can provide only partial insights and non-robust indications on the usefulness of the advisories (given unrealistic or not convincing assumptions on relevant trigger probabilities and population exposed to flooding). So far, some parts of your analysis cannot convincingly support a few central points of your conclusions. Thus, major revisions are recommended.

We thank the reviewer for their supportive appraisal of our work and their constructive and insightful feedback.

In response to these comments and suggestions we plan significant additional analysis for a revised version of the manuscript. In particular we will use new datasets created by the Kenya Red Cross Society (KRCS), which have been generated since submission of the manuscript.

Firstly we will use ward-level data on population exposed to flood risk instead of total population in order to improve estimates of the scale of preparedness implied by each advisory.

Secondly we will use a county-level database of all reported flooding between 2015-2019 to complement the verification of advisories against EM-DAT. This new dataset contains 184 unique days with reported flooding, which will enable us to evaluate skill statistics more robustly.

Description of additional changes and our responses follow.

**Major comments**

**1. The first major concern is that to estimate the relative scale of preparedness implied by each advisory, the population exposed and vulnerable to heavy rainfall and consequent impacts (flash floods, water-logging or riverine floods) should be used instead of the total population for each warned county. The total population living in a warned area seems an oversimplified and unrealistic proxy indicator, that does not provide a measure of the number of people likely to benefit from flood preparedness actions in the region and does not allow a comparison of the extent of preparedness action required between advisories. The authors partly recognise this issue, but do not address it properly and do not convince the reader on the value of their 'first-guess' estimates. A proxy estimate based on the total population per county does not seem a sensible approach even to provide a broad indication of the relative amount of preparedness appropriate for each advisory (see L. 170-174). The broad indication derived from this proxy could deliver the wrong message (or maybe the right message but for the wrong reasons), being based on assumptions that are not necessarily true (i.e. there is some correlation between total population of a county and vulnerable/exposed population to heavy rainfall per county, but the distribution of vulnerable population across counties may not match the distribution of total population).**

**Related to this, it should be also acknowledged that although there is an attempt at overlaying population density and rainfall accumulation observed over each advisory window (L. 262-265 and Figure 6b), in many cases the population living in an area receiving heavy rainfall does not coincide with the population potentially affected by flooding, especially for riverine flooding events. Please consider using some additional datasets of population potentially affected by flooding. For example, you might want to use some datasets that exist to better estimate potentially affected population by flooding at least in some areas, based on data available for past events in some regions of Kenya, at least to provide a case-study example of the extent of preparedness implied by a single or few advisories, e.g. for eastern Kenya you could use the dataset available in the OCHA's Humanitarian Data Exchange (HDX) platform, based on Sentinel-1 imagery acquired on 2018 : https://data.humdata.org/dataset/potentiallyaffected-population-by-flooding-in-eastern-kenya-2804**

We agree that the use of total population as a proxy for the scale of preparedness implied by advisories is an oversimplification. The analysis referred to intended here to provide a relative scale of preparedness, which it does to some extent (i.e. an advisory warning many small densely-populated counties might trigger a large intervention than one warning only a few large sparsely-populated counties).

However the reviewer is correct to highlight potential discrepancies in the distribution of the population exposed to flood risk with the distribution in general. For this reason we will update this

analysis with a dataset provided by KRCS, who have recently carried out analysis exposure to riverine flooding at ward level, as part of the IARP project. In particular we will use the data for population exposed to a 5 year return period flood; a 5 year event is the focus of the development of flood preparedness triggers in IARP. The data itself has been created by KRCS by integrating inundation areas estimated by ECMWF using GLoFAS with ward level population data.

Using flood exposure data will provide a much more realistic estimate of the scale of potential intervention implied by each advisory. We note that it will only consider preparedness actions aimed at the population exposed to riverine flood and not those exposed to flash flooding or landslides. We will add a discussion on this point.

Regarding the second point (the assumption that an action will be perceived as worthy in a location only if heavy rainfall falls on that location directly). The reviewer rightly points out that there can be a mismatch between local rainfall and flooding when rainfall falls upstream in catchment and floods lower reaches. This means that flooding can occur in a region which saw no heavy rainfall and so flood preparedness may still be perceived as worthwhile, despite no local rainfall. The implication of this is that the analysis of 'action worthiness' (figure 6b) should be interpreted conservatively. That is, the assumption we make does not capture flooding related to non-local rainfall and so the estimate should be considered as a lower bound on 'worthiness'; inclusion of flooding related to non-local rainfall would only increase the estimate. We note that explicitly estimating the contribution from non-local rainfall would require significant hydrological analysis which we consider out-of-scope - for the purpose of the analysis set out in the manuscript a lower bound estimate of worthiness is acceptable. We will add discussion of this point to the manuscript.

**2. A second major concern is that triggers should be defined more realistically, based on some relevant rainfall thresholds and effective probability triggers. You argue that you may avoid considering any specific rainfall threshold or probability because it would not 'provide robust statistics and precludes any meaningful statement' (e.g. see lines 138-140). However, while I agree that the sample size and the inconsistencies of the data from these 33 HRAs preclude any meaningful calculation of some verification metrics such as the reliability of probabilities, I think that the available probabilities and rainfall data could still be used to answer the questions in your paper more quantitatively. In other words, I agree that you cannot compare warnings with different levels of spatial aggregation, different temporal windows for accumulation of rainfall, etc., but you can still test whether the HRAs were useful overall for forecast-based early action based on some minimum quantitative analysis. For example, you can set a minimum rainfall threshold (maybe dependent on the window of accumulation, or a minimum with a larger window) and some significant probability based on the classes available, e.g. probability of heavy rainfall ¿ 33% (and not just above zero). The main problem I see in your analysis is that you have defined the action trigger as the probability (of heavy rainfall) exceeding zero, but an action trigger with a very low probability of unspecified heavy rainfall level seems a very unrealistic trigger even for low-regret actions. Such an approach is likely to lead to overconfident verification results on the value of the HRAs. For example, would a probability lower than 10% of heavy rainfall expected to fall over a big county still lead to any concrete action by government or humanitarian agencies? If you have valid reasons to think so and use a very low trigger probability, you should at least extend the discussion on this point to convince the reader of the validity of this analysis. Maybe you could support this**

choice based on some literature, or any reported practice in the humanitarian sector, explaining how this would be useful and what actions would be informed – otherwise all the analysis and conclusions seem to be based on unrealistic assumptions. Still, I believe that the analysis would be more valuable if you could show the performance of the warnings for significant probability levels and considering some meaningful rainfall thresholds that are more likely to be used as triggers.

If you defined triggers in a specific and realistic way, this would allow to make your analysis more concrete and link it to some specific actions to determine the extent to which the KMD HRAs could guide 'worthy' preparedness activity. Your definition of 'worthy' action seems being kept purposely vague (in line with a zero-probability trigger) and not clearly defined, referring to any preparedness assistance and no particular action (e.g. line 181). Using some specific examples of actions and trying to quantify whether taking these actions would be worth would be a natural step forward to give more concrete value to your analysis. Please consider including some specific action-based analysis or examples that could give more value to the article.

Assessment of advisories based on specific probability or rainfall amount thresholds is a reasonable approach, especially for evaluation of a potential FbF system which may benefit from the flexibility of a choice of trigger thresholds. However in this particular case there are issues which we believe make an evaluation in this way unreliable.

Firstly the majority of advisories (24 of 33) have the probability category 33-66% assigned. Only two advisories show probabilities below this (0-33%; $B$ and $K$). Using an increasing threshold above 0% first excludes these two (to which the overall conclusions are insensitive), following this the next exclusion would remove the largest category, leaving only seven (that is, requiring probabilities of greater than 66%). This seven are all in the early period and are not particularly accurate; as we conclude, this is may be related to improvement of the system over time. Thus sub-setting the verification to these 'high-probability' alone is not a meaningful indication of the potential skill of a future FbF system based on the advisories.

Moreover it is unclear that the choice of the probability values is based on objective estimation of risk. Table 1 demonstrates that a variety of probability thresholds were used in the early years of the advisories, however from 2018 every single advisory used 33-66%. This suggests that this broad category may be used 'by default' rather than arising from objective consideration of the risk of heavy rainfall during each specific event. Thus, it is not clear that meaningful information on risk is contained in these probabilities. In addition, it is important that the the forecast upon which an FbF system relies is be as objective as possible. If a large release of money were dependent on the exact value of an entirely subjectively-determined probability (for example), there is a non-scientific incentive to modify the probability. So, we feel that a trigger of 'release of an advisory' is more insensitive to this potential issue than a trigger of a specific probability threshold.

Regarding the use of rainfall thresholds to sub-select the advisories: this runs into the problem of variations in length of the advisory window (mentioned in section 2.2 and by the reviewer above). So whilst mention of 50mm in the advisory could be used as a threshold, these advisories can warn of quite different events; some warn of 50mm on a specific day, others expect 50mm accumulation over a period of four days. In addition some advisories targeting multi-day windows warn that 50mm in 24 hours is expected on one of the days in the period, whilst others only warn of 50mm total over

the four days. Using a criteria of 'mentions 50mm' would also exclude advisories which mentioning smaller totals but are focused on shorter windows. These shorter more intense rainfall events could be just as damaging as higher accumulation over a longer window. Given then the 'problem' of variable window length, and the inconsistent definition of accumulation within each advisory (per 24 hours, or total accumulation), we do not think it is insightful to use these thresholds to subset the data.

These issues around using probability and accumulation thresholds to subset the verification are described in the paper (line 108-9, section 2.2), so we do not feel this requires additional detail. However we do note the reviewers point:

*"would a probability lower than 10% of heavy rainfall expected to fall over a big county still lead to any concrete action by government or humanitarian agencies?".*

We have essentially suggested this in line 140, by saying that:

*the action trigger is defined as the probability of heavy rainfall (of any specific threshold) exceeding zero..*

However on reflection we feel this is somewhat misleading, since very low probabilities have not been a feature of the advisories for several years, and the information content of the specific values of the probabilities is unclear (see discussion above). Instead we would frame action as being triggered not on 'non-zero probability', but on 'the decision to release an advisory'. We will modify this line in a revised manuscript.

Finally regarding the suggestion of evaluating the worthiness of specific actions. This would require estimates of the cost and potentially avoided losses associated with certain actions. We are not in possession of this information and our estimates would be highly uncertain and likely misleading. A full econometric analysis of an FbF system would require significant additional research and we consider this beyond the aims of the current paper (e.g. see the research questions outlined in lines 146-9). However we would discuss this need in a revised manuscript.

**3. There is a lack of evaluation of misses (missed events in the warnings) or at least a discussion on it: the evaluation of observed events is only based on seven reported impactful events (the most significant floods events in the EM-DAT dataset), and there is no proper evaluation of 'misses' and 'hit rates' based on a large sample, which is a limitation of the data and period available. Despite the obvious sample size issues, it would be probably possible to include in the analysis in Section 3.1/Section 3.3 some more information on observed events also based on other data (not only EM-DAT). Are there any other significant flood events beyond the 7 events from EM-DAT (e.g. maybe events with less than 10 fatalities but still high number of affected people / households affected or damaged) that have been missed by the HRAs during the study period? Please discuss this.**

**If more events were available (beyond EM-DAT), in Section 3.1 you could include an evaluation of hit rates per county using CHIRPS as reference and/or in Section 3.3 a proper evaluation of misses based on the larger sample of reported impactful events. Without a full evaluation of hits and misses, the 'hits' picture that is given**

might be misleading and uncomplete. You mentioned some misses because related to the time windows of advisories issued (e.g. for advisories warning "wrong" counties, see lines 233-234). It would be useful for decision makers if you could calculate a proper hit-rate even if based on a sample of 33 advisories. You could focus on a specific trigger probability and threshold rainfall, for example you could keep the 50 mm nominal threshold case and use a specific probability threshold. Of course, using only a single rainfall threshold across a big country as Kenya is not a proper location-specific indicator of flood impacts, but this would be still useful. Additional analysis with some more observed events (if you had more than these 7) would probably help understand also whether the step change in the advisories in 2017 (access to GHM) reduced the number of misses, as it seems the case from your analysis based on 7 events (Figure 7) and might be expected from the increasing number of advisories per year (Fig. 2a). Section 3.3 could then be more complete by focusing on both hits and misses, by reporting how many observed events were not preceded by advisories.

Since preparation of the manuscript, KRCS have carried out work to identify all reported flooding events in Kenya. We have secured the use of this dataset for inclusion in a revised analysis. The conditions for inclusion in this new dataset are less strict than EM-DAT and so many more events are included: for our study period of 2015-2019 the dataset contains a total of 184 unique days of recorded flood events. We will use this data to carry out additional analysis of hits and misses as is suggested. In response to the question of using specific probability thresholds, please see our response to comment 2 above.

4. The analysis provides useful insights on the possible limitations of the HRAs and reccomendations, but the discussion should make more efforts in understanding and explaining the current limitations of the HRAs. This is essential to provide more specific recommendations for improvement of the HRAs. One of the major limitations of the HRAs that arises from the analysis is the lack of precise spatial information in the warnings beyond the county-level information. As you suggest, free-shape warning areas should be used instead of administrative county boundaries. However, you also explain that such free-shape warning polygons are currently generated by the GHM and are already in use at KMD. Thus, "KRCS could then overlay these with maps of population exposure and vulnerability to flood risk, in order to further narrow down targets for intervention". Then why more precise warnings are not issued yet? It is not clear what is the current bottleneck for providing more spatial detail in the advisories, and it would be important to understand whether there are either scientific / technical or institutional / economical barriers that prevent this, e.g. either whether it's a lack of resources (e.g. GIS technicians) at KMD, or whether there has not been enough co-production effort made so far to enable a full use of the GHM information at KMD, or whether there are some information layers missing (e.g. flood extent maps which do not coincide with heavy rainfall extent). Without an extensive discussion on this point, it is not clear how the spatial detail in the advisories could be improved.

To understand why precise warnings are not issued yet one must consider the primary purpose of the advisories: to provide broad-scale warnings of future potential adverse weather to multiple sectors and the public. The text-based county-level warnings from the advisories are fit for this purpose and as such there is no internal drive from KMD to add additional detail. Only when it has been suggested to use the advisories for a purpose requiring spatial precision (i.e. within

an early action protocol), does the motivation for development arise. To our knowledge, our work voices this need for the first time in the literature.

In addition, the Met Office GHM remains a prototype tool and has only been introduced to KMD over the past few years to assess its suitability for enhancing and supporting the development of impact-based warnings. Given that the GHM focuses on forecasting high-impact events, it takes time to both assess the usefulness of the tool but also its potential value in supporting KMD advisory development. One might not expect it to prompt a change in the format of the advisories in this time.

Informal discussion suggests no technical or economic barrier to adding detail to the advisories, however, there may reasonable resistance to modifying the format of the advisories, related to dissemination. In the current form, a bare minimum of technical knowledge is required to correctly interpret the information which facilities understanding and easy dissemination (e.g. through local radio, in-person broadcasts to local communities). To add additional information may limit the ease with which they are disseminated, as well as their interpretability (for instance if some of the audience are not familiar with reading data from a map). Getting the balance between interpretability and detail right is a challenge for KMD (and Met Services in general). We will add some comment on these points in a revised version of the manuscript.

**5. Finally, it would be essential to discuss in this article how the HRAs fit within the need "for strengthened coproduction of forecast information and products" (now widely recognised as you also recognise). Is there any issue of national ownership in the use of GHMs from the UK MetOffice in the HRAs?**

**From your article, it seems that only the subjective interpretation of the forecasts and the writing of the HRAs summary is carried out within KMD and so within the mandated agency in Kenya. How is this perceived at the national level? I can see that the actual sources of forecast information in the HRAs are not mentioned in the HRAs (see Figure 1, no field on 'data sources') so maybe there is no general perception from the communities involved in Kenya or even from county directors / national policy makers around that issue. However, this point would need attention in such a paper.**

**Also, it would be useful to detail whether KMD get access to raw forecast data from ECMWF and UK MetOffice or only to some end-product maps and the level of spatial detail in these maps (e.g. do KMD get any shapefiles or netcdfs data? At which resolution?). This point might help explain one of the major current limitations (lack of spatial detail in the HRAs) and how KMD and their international partners could deal with it to improve the advisories.**

To address these points in turn (detail will be added to the manuscript):

Strengthened coproduction is certainly important; part of the co-production process involves feedback to forecast producers on the suitability of products for a particular purpose. The work we describe here can form part of this process.

KMD use forecast inputs from many global producing centres in the production of their forecasts; use of the GHM in addition to these is not perceived to lead to issues of national ownership.

GHM is made available to KMD via a webpage, which they can review and consider in the context of other model data that is available to them. The layers are published as WMS (Web Map Service) layers which can be ingested into any compatible geospatial software for onward analysis, however this mechanism of data access has not been tested with KMD. If there was interest from KMD in receiving the forecasts in a different format (i.e. as shapefiles or NetCDF) then this could be arranged, or alternatively training on accessing the WMS layers could also be provided (dependent on software availability).

**Specific comments - L. 34: it would be good to specify how this UK-funded project fits within the local context and is linked with other projects mentioned (IARP) or not; are these projects making some efforts in coproduction and capacity building and how specifically (only by giving the outputs of GHMs models to local agencies or is there something more, e.g. capacity building efforts)? It would sound very sensible to discuss this.**

ForPAc has worked to investigate the possibility for forecast-based action across several case studies and timescales. These includes seasonal forecasting for drought events, urban flooding in Nairobi and extending the Nzoia flood early warning system using subseasonal forecasts. The Kenya/UK team has worked closely with mandated agencies and has undertaken significant engagement with stakeholders, including participatory impact pathway analysis, climate information training workshops and development of forecast production methodology at KMD and ICPAC. Given that the project is only mentioned in passing during the introduction, we will add a short extra line to describe the work.

**- L. 45: do KMD work on their own on hydro-meteorological forecasting? The mandates and institutions involved in hydrological warnings in Kenya should be clarified, as KMD is the meteorological agency, and there is also a national hydrological agency, the Water Resources Authority (WRA, https://wra.go.ke/) that should be responsible of flood forecasting activities alongside KMD (e.g. see FLOOD ADVISORIES issued by WRA; see also ODI working paper 553, April 2019, "Reducing flood impacts through forecast-based action" by Lena Weingärtner et al.). The links between KMD and WRA are not clear nor mentioned in the paper. It seems an important point to discuss, as probably a closer collaboration with forecasters at WRA could help make the HRAs by KMD more precise, with impact-based focus and hydrologically meaningful. Is there a link between the KMD Flood Forecasting Unit and WRA? Or is this a current institutional barrier? Hydrological forecasting is at the interface between met and hydro agencies not only in Kenya but in many countries and similar questions may arise elsewhere. Some more context about this important point should be provided.**

This is a particularly insightful comment. Work on ForPAc has established that the institutional links between KMD and WRA could be stronger. Efforts are underway which will address this, notably a national-scale flood forecasting system is in development. The reviewer is likely correct that closer collaboration with WRA could help make the HRA more precise, and we will add this suggestion to the discussion.

**- L. 101: I would suggest specifying "The GHM system", to avoid confusion with other possible meanings, e.g. HRA or other systems just mentioned in the text - Section 2.2**

**(Verification Approach): L. 150-219 are difficult to follow and should be reorganised in a more clear or compact way (e.g. with bullet points for all the methods and data associated to each question). - L. 200-206: the part about the dam collapse of May 2018 seems excessively long in the context of this section and should be kept more concise; as it stands, that part does not flow well within the paper.**

We will make these changes in a revised version of the manuscript.

**- L. 243-244: "As these advisories associate each warning with a probability, these findings are quite consistent" – this sentence is obvious and not specific enough. You could try to add something more relevant and specific, as the previous remarks in terms of convective rainfall and percentage of warned area are interesting. For example, could you see visually any link between percentages of area warned which receive rainfall accumulation above the 50mm threshold and geographical locations/counties that are known to be more subject to convective rainfall?**

With this sentence we only mean to highlight that although a lack of heavy rainfall in a location would not necessarily invalidate a probabilistic forecast. This may be obvious to some, but we feel it is worth stating explicitly here. The suggestion that 50mm warnings may be more frequent in convective regions is interesting, although it is difficult to confirm this: visual inspection shows that nearly all counties feature in at least one 50mm advisory.

**- L. 347: "These kinds of actions would have significant costs, so more than ten triggers in a year may not be realistic" – that sounds reasonable but too approximate to be stated in this way, could you provide more details (e.g. approximate estimate of costs and resources) and improve the sentence? Are there any references supporting this sentence (and the number of ten triggers)? Are there any estimates in the scientific/economic literature or in reports of humanitarian agencies on the resources that would be needed / are available for early action and flood preparedness in Kenya or maybe in any specific region within the country?**

From the Red Cross perspective, the typical event to be targeted is an extreme event, rather than one which may occur every year. See for example step 6 in the FbF practitioners handbook (https://manual.forecast-based-financing.org/chapter/set-the-trigger/). In addition, the developing KRCS protocol for flood preparedness is focusing first on a one in five year return period flood event. However, it is not possible to say for sure that ten triggers is not realistic; there may be low-regret actions for flood preparedness (such as fast-tracking drainage clearance which has already been planned and budgeted for). We will add these details to the revised manuscript.

**- L. 405: "For this purpose they are effective" sounds a bit overstated, e.g. given the lack of spatial precision that you highlighted in the paper, the large area warned by identifying counties in the text may not be effective (people in an affected county may not take county-scale warnings so seriously, if these are preceded by warnings that were not followed by any event in their specific area in that warned county).**

We disagree; we state that they are effective at alerting to the possibility of heavy rainfall and believe that do this successfully. Of course the warnings may not be sufficient to guide specific preventative actions. It is our understanding that the advisories are an important information

source followed by humanitarian agencies (e.g. Kilavi et al. (2018) note dissemination and use of HRA during the long rains).

**- "Data and verification approach" section: some final parts do not flow well and could be improved (e.g. more clear organisation by points and questions addressed); some parts need more details or references to the scientific literature or humanitarian reports to back-up some assumptions made (e.g. see also remarks in major points above).**

We will work to improve the flow, details and references of this section in a revised version of the manuscript.

**- "Discussion and recommendations" section: there is a lack of discussion on the temporal consistency in the HRA dataset. There is a difference in the source rainfall forecast in the new data in recent years, but also the number of HRA has increased. So, what role does this inconsistency play in comparing earlier years with more recent ones? For example, the number of hits in recent years is expected to be higher simply because there are more warnings issued.**

This is a reasonable point; more frequent issuance could lead to more hits. However we do also note that earlier advisories tended to perform more poorly (see figures 4 & 6), which suggests that the quality of the advisories has improved, along with their frequency. We will add this discussion to the revised manuscript.

**Technical corrections - Table 1, column 3 - "Period length" is missing the units (days, probably)? - L. 2 (Abstract): "Forecast-based Action/Finance (FbA)", it seems that Forecast-based Financing is more used than Finance, please check. - L. 19: Climate risk or better hydro-meteorological risk? - L. 27: repetition to avoid in (see Wilkinson, for. . .) as L.23 already mentioned it - L. 29: 'individual expenses' and/or probably even more 'community expenses'? - L. 164-165: it's fine to focus the discussion on results for 50mm accumulation, but maybe you can say here more explicitly that you took this threshold as "working definition for heavy rainfall", as sometimes later in the results section this is the wording you use (e.g. Line 246), so good to define it clearly from the methods, still mentioning the limitations of it as you do. - L. 167: "To answer question .. we estimate" is missing the question number - L. 399-400: see sentence "We find that an increase in skill over time, and that. . .", to be corrected. - L. 438: higher-cost actions -L. 456: repetition, "would would" - L. 470-471: repetition of "in Kenya" - L. 484: "mitigating the risk from risks", I would avoid the repetition - Figure 4a, caption: please clarify whether by "inner and outer quartiles" you mean "inner and outer fences" (which seems more common wording in this context)? – see "(dark/light shading shows inner/outer quartiles and dot indicates the median". (by the way, a parenthesis is missing there).**

These will be fixed in the new version of the manuscript.

---

## Author Comment (AC2) · 26 Aug 2020

**NHESS-2020-122: Are Kenya Meteorological Department heavy rainfall advisories useful for forecast-based early action and early preparedness?**

**Response to review RC2**

Original comments are duplicated below **in blue**, with our responses following in turn.

**This short paper gives an insight into how flood warnings are generated at the Kenyan weather service and how their skill evolved over the last 5 years. Despite the relatively small number of cases and some data inhomogeneity, I find the paper useful for practitioners and generally welcome publication of such work. Overall the paper is well written and logically structured. There is, however, substantial room for improvement with respect to data and the evaluation methodology as detailed in the following major and minor comments.**

We thank the reviewer for their support of our work and their useful feedback. In order to meet the request for improved data and evaluation we plan significant additional analysis for a revised version of the manuscript. In particular we will use new datasets created by the Kenya Red Cross Society (KRCS), which have been generated since submission of the manuscript.

Firstly we will use ward-level data on population exposed to flood risk instead of total population in order to improve estimates of the scale of preparedness implied by each advisory.

Secondly we will use a county-level database of all reported flooding between 2015-2019 to complement the verification of advisories against EM-DAT. This new dataset contains 184 unique days with reported flooding, which will enable us to evaluate skill statistics more robustly.

A full description of additional planned changes and our responses follow.

**MAJOR COMMENTS:**

**1.) Evaluation procedure: Classically one would consider hits, false alarms, missed events and correct non-events. This would enable the computation of all the classical scores such as Proportion Correct, Heidke Skill Score etc. Your analysis gives a good idea of hits and false alarms but the missed events are only treated with respect to the 7 flood cases from the EM-DAT database. Can you not use CHIRPS to give some idea for missed heavy precip events that you could define to have a certain intensity and spatial reach (as pointed out in Point 2 of Reviewer 1)? After that, all days that remain would be correct negatives. This would allow a more quantitative treatment of skill.**

Since preparation of the manuscript, KRCS have carried out work to identify all reported flooding events in Kenya. We have secured the use of this dataset for inclusion in a revised analysis. The conditions for inclusion of flood data are less strict than EM-DAT and so many more events are included: for our study period the dataset contains a total of 184 unique days of recorded flood

events. We will use this data to carry out additional analysis of hits and misses as is suggested.

In response to the question of using CHIRPS to define misses; this faces a challenge in defining a consistent event across advisories (which vary in forecast window) and across counties (which vary considerably in size). This is discussed in the manuscript in section 2.2. We had previously investigated the possibility of requiring a certain intensity and spatial reach in observations (e.g. requiring a certain area to receive significant accumulation). However the definition is somewhat arbitrary and different choices of event definition lead to quite different results, without a clear way to objectively choose between them. For this reason, we instead have used the observed record of flooding to define hits as it is less ambiguous. Given the extended flood database we mention above, we will now be able to more robustly calculate the hits and misses as suggested above.

**2.) Language: Overall the paper is nicely written and the level of language high. However, some passages are a bit wordy and redundant and I would therefore ask the authors to careful assess the potential for shortening. Given your overall low levels of statistical significance, I would also be a little more cautious with statements on skill throughout the text.**

We will reduce redundant text and ensure our statements on skill are consistent with the results presented in a revised version of the manuscript.

**3.) Abstract: In its current state the abstract does not really explain well what the paper is all about and in what way it is important, new and special. There should be more information on data, method, results and limitations.**

We will increase the information content in the abstract in a revised version of the manuscript.

**4.) Rainfall data: This is always an issue. There are many different products with strengths and weaknesses. Please provide more evidence that CHIRPS is a good one (the best?) to use and possibly repeat exercise with an alternative source of information.**

We use CHIRPS as it takes advantage of satellite coverage, whilst 'ground-truthing' against station records. CHIRPS compares favourably to other satellite-based datasets over East Africa (Dinku et al 2018).

In addition, a global evaluation of 22 rainfall datasets (Beck et al 2017) recommend the use of CHIRPS in particular for tropical regions. Beck et al 2017 note difficulty in providing reliable recommendations in regions such as Africa where rain gauge data is limited. However for Kenya in particular the station density used in CHIRPS is relatively good (see for example `https://data.chc.ucsb.edu/products/CHIRPS-2.0/diagnostics/chirps-n-stations_byCountry/Kenya/Kenya.1981.01.png`).

Weaknesses with CHIRPS include spurious drizzle and an underestimation of peak magnitudes (Beck et al 2017). Our analysis is likely to be insensitive to spontaneous drizzle, although an underestimate of peak rainfall implies a conservative bias to our evaluation of the skill of advisories for predicting threshold accumulation (e.g. figure 4b, figure 6). Here the advisory area receiving threshold accumulation may be higher than CHIRPS suggests. Having said this, Beck et al 2017 find

[Figure]

[Figure]

Figure 1: 99.9th percentile of CHIRPS daily rainfall over Kenya, from Climate Explorer

that the most extreme rainfall (e.g. 99.9 percentile of daily rainfall) shows most underestimation. Analysis shows that a 99.9 event over Kenya in CHIRPS ranges from 30-130mm per day (figure above), whilst our focus is on multi-day accumulations of 25, 50, 75 and 100mm, suggesting that CHIRPS estimates of totals close to our thresholds of interest are less affected by underestimation compared to the highest magnitudes.

Overall we will add a justification of the use of CHIRPS in the manuscript and following the above. However do not feel that the use of an additional precipitation dataset for verification would bring more robust results. In particular, our analysis ultimately moves beyond uncertainty in rainfall observations by focusing directly on specific flood events (EM-DAT, and the additional flood record from KRCS, see point one above).

Beck, H.E., Vergopolan, N., Pan, M., Levizzani, V., Van Dijk, A.I., Weedon, G.P., Brocca, L., Pappenberger, F., Huffman, G.J. and Wood, E.F., 2017. Global-scale evaluation of 22 precipitation datasets using gauge observations and hydrological modeling. Hydrology and Earth System Sciences, 21(12), pp.6201-6217.

Dinku, T., Funk, C., Peterson, P., Maidment, R., Tadesse, T., Gadain, H. and Ceccato, P., 2018. Validation of the CHIRPS satellite rainfall estimates over eastern Africa. Quarterly Journal of the Royal Meteorological Society, 144, pp.292-312.

**5.) Section 2.2: I think that the approach you are taking is largely well conceived (but note my reservations under Point 1) given all the restrictions at hand but the section as written is quite long and your quantitative metrics are only described and nowhere cast into formulas. I suggest giving this section a clearer structure and a more "recipe like" description of how you compute metrics. If you give names or abbreviations to your metrics, you would not need to repeat the description again in Section 3.**

We will review this section in a revised version of the manuscript and attempt to reduce unnecessary detail. However we do feel that given the particular challenges to verification, some space is needed in the manuscript in order to motivate and justify why we are unable to follow a standard approach. We are not convinced it will aid readability to introduce equations to abbreviate the metrics we use.

**6.) EM-DAT: I find the thresholds of 10 deaths too high and would feel that even one death would justify a weather warning. Given that you have authors from Kenya that may have access to government documents, is there no alternative source of information that would give you a list of flood events of smaller magnitude, too? This would much improve your statistics relative to the few events in EM-DAT!!**

See point one above: we plan to evaluate the advisories against a larger record of flood events provided by KRCS.

**7.) Population numbers: I agree with Reviewer 1 that a distinction between all population of a county and the fraction likely affected by floods (in particular riverine) would be desirable. However, I can imagine that such fractions are not easily available and feel that the paper would be of value without it. In this case the authors could raise this point more clearly in the text and give at least some orders of magnitude from literature.**

Given this comment and the request from Reviewer 1, we plan to improve this part of the analysis by using a dataset provided by KRCS, who have recently carried out analysis of exposure to riverine flooding at ward level as part of the IARP project. In particular we will use the data for population exposed to a 5 year return period flood (a 5 year event is the focus of the development of flood preparedness triggers in IARP). The data itself has been created by KRCS by integrating inundation areas estimated by ECMWF using GLoFAS with ward level population data.

Using flood exposure data will provide a much more realistic estimate of the scale of potential intervention implied by each advisory. We note that it will only consider preparedness actions aimed at the population exposed to riverine flood and not those exposed to flash flooding or landslides. We will add a discussion on this point.

**MINOR COMMENTS: 1.) Punctuation: There are a lot of places with inconsistent or suboptimal use of commas. Please check carefully throughout the entire manuscript. 2.) L2: remove "a" as in plural 3.) L5-6: What are you trying to say with this sentence. Please reword! 4.) L12: no comma 5.) L19: is it really a "movement"? In L31 it is called a "society"? 6.) L30: IFRC? 7.) Section 1: this gives a nice introduction to the topic but some bits are a little redundant and could be streamlined. 8.) L75-76: avoid repetition of "improve" 9.) L120: remove period after figure 2 10.) L125: better turn this into a proper sentence 11.) L167: this question? 12.) L194: requires? 13.) Table 1: Why don't you merge the first two entrances? 14.) L234: fell during . . . 15.) L241: "quite a reasonable chance" is very fuzzy, reword! 16.) L245-248: What result or figure does this paragraph refer to? 17.) Figure 4 could be discussed in a little more detail. 18.) Figure 5 I would rather include in the Methods section 2. You can then also discuss there the difference between all people and those affected by a given flood (see above). 19.) L255: remove "extreme" as upper bound is already an extreme 20.)**

**L286: highest number? 21.) L319: on 18th November? 22.) L385: I would maybe not use the word "all" here, as it remains a probabilistic problem, where some missed events are unavoidable. 23.) L441: double period 24.) L443: comma instead of period 25.) L456: 2x would 26.) L458-59: not a proper sentence 27.) Section 4.2.2: Too much detail to my taste. This is a scientific paper and not a government technical document. 28.) Figure 2 caption: include that these statistics are done for the cases listed in Table 2. 29.) Figure 4 caption: these should be 5kmx5km gridpoints 30.) Figure 5 caption: two brackets at end 31.) Figs.: I would generally not start a caption with a question.**

These will all be addressed in a revised version of the manuscript. Though regarding too much detail in section 4.2.2 (point 27), we feel that provides the broader institutional context of developing hazard early warning systems in Kenya and will be of interest to (at least some) readers of the work, and fits within the scope of NHESS.

---

## Author Response (AR2)

**NHESS-2020-122: Are Kenya Meteorological Department heavy rainfall advisories useful for forecast-based early action and early preparedness?**

**Cover letter**

Thank you for the positive review of the paper. We present our revised version following final technical comments from the second reviewer. Point responses are given below. I have also added the University of Bristol to my affiliation as I moved institutions over the summer and carried out the revisions here.

**David MacLeod (on behalf of co-authors)**

*MINOR COMMENTS:*

*1.) L46: 2x "the"*

Fixed.

*2.) L151: remove second "that"* This has been fixed.

*3.) L235ff: sentence long and hard to read*

We have split up this sentence and clarified.

*4.) L417: remove space before period*

This is not an error in the LaTeX source; rather the compilation is breaking the line after the citation "(see figure 3 and Kilavi et al 2018). This will hopefully be fixed when formatted by the journal.

*5.) L431: better "the skill analysis"*

We have added added "skill" before analysis here as suggested.

*6.) L470: "... on extreme events rather than floods occurring every year (RCRCCC 2020)"*

We have made the suggested ammendment.

*7.) L491: comma before respectively*

Comma has been added.

*8.) L504: hit rate*

This has been fixed.

*9.) L570: comma before "although"*

A comma has been added.

*10.) L587: short-range*

This has been changed.

[revised manuscript text omitted]